# Informed Initialization for Bayesian Optimization and Active Learning

**Carl Hvarfner**
Meta
hvarfner@meta.com

**David Eriksson**
Meta
deriksson@meta.com

**Eytan Bakshy**
Meta
ebakshy@meta.com

**Max Balandat**
Meta
balandat@meta.com

## Abstract

Bayesian Optimization is a widely used method for optimizing expensive black-box functions, relying on probabilistic surrogate models such as Gaussian Processes. The quality of the surrogate model is crucial for good optimization performance, especially in the few-shot setting where only a small number of batches of points can be evaluated. In this setting, the initialization plays a critical role in shaping the surrogate's predictive quality and guiding subsequent optimization. Despite this, practitioners typically rely on (quasi-)random designs to cover the input space. However, such approaches neglect two key factors: (a) space-filling designs may not be desirable to reduce predictive uncertainty, and (b) efficient hyperparameter learning during initialization is essential for high-quality prediction, which may conflict with space-filling designs. To address these limitations, we propose Hyperparameter-Informed Predictive Exploration (HIPE), a novel acquisition strategy that balances predictive uncertainty reduction with hyperparameter learning using information-theoretic principles. We derive a closed-form expression for HIPE in the Gaussian Process setting and demonstrate its effectiveness through extensive experiments in active learning and few-shot BO. Our results show that HIPE outperforms standard initialization strategies in terms of predictive accuracy, hyperparameter identification, and subsequent optimization performance, particularly in large-batch, few-shot settings relevant to many real-world Bayesian Optimization applications.

## 1 Introduction

Bayesian Optimization (BO) (Frazier, 2018; Garnett, 2023; Jones et al., 1998; Mockus et al., 1978) is a principled framework for sample-efficient global optimization of black-box functions with applications across diverse fields such as biological discovery (Griffiths and Hernández-Lobato, 2020; Stanton et al., 2022), materials science (Ament et al., 2023b; Attia et al., 2020; Frazier and Wang, 2016), online A/B testing (Agarwal et al., 2018; Feng et al., 2025; Letham et al., 2019), and machine learning hyperparameter optimization (HPO) (Feurer et al., 2015; Snoek et al., 2014). BO combines a probabilistic surrogate model—commonly a Gaussian Process (GP)—with an acquisition function to select where to evaluate the unknown objective function. In many applications, the runtime of a single black-box function evaluation may restrict the experimenter to a small number of *batches* – the number of sequential rounds of experiments – but many real-world experimental setups permit conducting multiple experiment simultaneously (e.g., on a parallel compute cluster, in a randomized controlled trial, or batch-testing multiple specimens in a lab).

39th Conference on Neural Information Processing Systems (NeurIPS 2025).

The success of Bayesian Optimization in practice is highly sensitive to the quality of the surrogate model (Eriksson and Jankowiak, 2021; Foldager et al., 2023; Hvarfner et al., 2023). This is a challenge particularly during the early stages of optimization when few observations are available. In these early stages, a small set of inputs is typically selected at random, or via space-filling or quasi-random sampling strategies such as Latin Hypercube Sampling (LHS) or scrambled Sobol' sequences (Bossek et al., 2020; Owen, 2023). While such strategies aim to achieve broad coverage of the input space for initialization of the surrogate model, they may not necessarily result in improving predictive accuracy, or reduce predictive variance, across the entire input space Ren and Sweet (2024). Moreover, they neglect a a second crucial aspect of modeling: the need to accurately infer the model's hyperparameters (Zhang et al., 2019), such as the kernel lengthscales of a GP. With accurate hyperparameter estimates, variation in unimportant dimensions will have less influence on the selection of points in subsequent iterations of BO, leading to more sample-efficient optimization (Eriksson and Jankowiak, 2021; Hvarfner et al., 2023; Müller et al., 2023). Conversely, poor hyperparameter estimation may cause subsequent BO iterations to fail to make meaningful progress (Berkenkamp et al., 2019), which may be caused by misidentifying signal for noise, exploring irrelevant dimensions, or returning poor terminal recommendations (Hvarfner et al., 2022).

In this work, we provide a principled approach to addressing this initialization challenge. Our main contributions are as follows:

1. We propose Hyperparameter-Informed Predictive Exploration (HIPE), an acquisition function for initialization that optimizes for both predictive uncertainty reduction and hyperparameter learning.

2. We derive a closed-form expression for this objective in the case of Gaussian Process models, and implement a practical Monte Carlo approximation to make it amenable to batched optimization.

3. We conduct extensive experiments in active learning and Bayesian Optimization on synthetic and real-world BO tasks, demonstrating that HIPE outperforms competing methods in terms of both model accuracy metrics and BO performance in few-shot, large-batch settings.

## 2   Background

### 2.1   Gaussian Processes

Gaussian Processes (GPs) are a widely used surrogate model in BO due to their flexibility, closed-form and well-calibrated predictive distributions. GPs define a distribution over functions, $\hat{f} \sim \mathcal{GP}(m(\cdot), k(\cdot, \cdot))$, specified by a mean function $m(\cdot)$ and a covariance (kernel) function $k(\cdot, \cdot)$. For a given location $\boldsymbol{x}$, the function value $\hat{f}(\boldsymbol{x})$ is normally distributed, with closed-form expressions for the predictive mean $\mu(\boldsymbol{x})$ and variance $\sigma^2(\boldsymbol{x})$. In practice, the mean function is often kept constant, leaving the covariance function to capture the structural properties of the objective.

To model differences in variable importance in GPs with stationary kernels, each input dimension is commonly scaled by a lengthscale hyperparameter $\ell_i$, a practice known as Automatic Relevance Determination (ARD) (Williams and Rasmussen, 1995). Additional, optional hyperparameters include a learnable noise variance $\sigma_\varepsilon^2$ and signal variance $\sigma_f^2$. The full set of hyperparameters $\boldsymbol{\theta} = \{\boldsymbol{\ell}, \sigma_\varepsilon^2, \sigma_f^2\}$ may be learned either by maximizing the marginal likelihood $p(\mathcal{D} \mid \boldsymbol{\theta})$ (Maximum Likelihood Estimation, MLE) or by incorporating hyperpriors $p(\boldsymbol{\theta})$ to perform Maximum A Posteriori (MAP) estimation. Alternatively, a fully Bayesian treatment (Lalchand and Rasmussen, 2020; Osborne, 2010) integrates over $\boldsymbol{\theta}$ to approximate the full Bayesian posterior distribution using Markov Chain Monte Carlo (MCMC) methods, thereby explicitly accounting for hyperparameter uncertainty. For additional background on GPs, see (Rasmussen and Williams, 2005).

### 2.2   Bayesian Optimization

Bayesian Optimization (BO) (Frazier, 2018) is a sample-efficient framework for finding the maximizer $\boldsymbol{x}_* = \arg\max_{\boldsymbol{x} \in \mathcal{X}} f(\boldsymbol{x})$ of a black-box function $f : \mathcal{X} \to \mathbb{R}$ over a $D$-dimensional input space $\mathcal{X} = [0, 1]^D$. The function $f$ is assumed to be expensive to evaluate and observable only through noisy point-wise measurements, $y(\boldsymbol{x}) = f(\boldsymbol{x}) + \varepsilon$, where $\varepsilon \sim \mathcal{N}(0, \sigma_\varepsilon^2)$.

At the core of BO is an *acquisition function*, which uses a surrogate model to quantify the (expected) utility of candidate points $\boldsymbol{x}$. Acquisition functions balance exploration and exploitation typically through greedy heuristics. Popular examples include Expected Improvement (EI) (Bull, 2011; Jones et al., 1998) and its numerically stable variant, `LogEI` (Ament et al., 2023a), as well as the Upper Confidence Bound (UCB) (Srinivas et al., 2012, 2010). In batch BO, multiple points are selected in parallel to accelerate data collection. This is often achieved by computing and jointly optimizing a (quasi-)MC estimate of the utility $u$ associated with acquisition function over the full batch $\boldsymbol{X} = \{\boldsymbol{x}_1, \boldsymbol{x}_2, ..., \boldsymbol{x}_q\}$ of size $q$ (Balandat et al., 2020; Wilson et al., 2017; Wilson et al., 2020).

### 2.3 Bayesian Active Learning

Bayesian Active Learning (BAL) and Bayesian Experimental Design (BED) (Chaloner and Verdinelli, 1995) aim to improve predictive models by selecting data points that are most informative, either with regard to the model or to future predictions. A key quantity is the Expected Information Gain (EIG):

$$\text{EIG}(\xi; y(\boldsymbol{x})) = \text{H}[\xi] - \mathbb{E}_{y(\boldsymbol{x})}\left[\text{H}[\xi|y(\boldsymbol{x})]\right], \tag{1}$$

where H is the Shannon (differential) entropy and $\xi$ is a parameter of interest. Importantly, EIG is symmetric, and can be equivalently formulated as an entropy reduction over $y(\boldsymbol{x})$, instead of $\xi$.

**Bayesian Active Learning by Disagreement** Bayesian Active Learning by Disagreement (BALD) (Houlsby et al., 2011; Kirsch et al., 2019) selects query points that maximize the mutual information between model predictions and hyperparameters $\boldsymbol{\theta}$:

$$\text{BALD}(\boldsymbol{x}) = \text{EIG}(y(\boldsymbol{x}); \boldsymbol{\theta}) = \text{H}[y(\boldsymbol{x})|\mathcal{D}] - \mathbb{E}_{\boldsymbol{\theta}}[\text{H}[y(\boldsymbol{x})|\mathcal{D}, \boldsymbol{\theta}]]. \tag{2}$$

BALD identifies locations where models within an ensemble exhibit the greatest disagreement in predictive uncertainty. In the GP setting, this often leads to axis-aligned queries when there is high uncertainty in the lengthscales, and to repeated queries when observation noise is highly uncertain. Notably, BALD is model-agnostic and can be applied to a wide range of surrogate models and hyperparameters, including subspace models (Garnett et al., 2014) and additive decompositions (Gardner et al., 2017; Hvarfner et al., 2023).

**Negative Integrated Posterior Variance** The Negative Integrated Posterior Variance (NIPV) (Seo et al., 2000) criterion selects queries that minimize the expected posterior variance over a test distribution $p_*(\boldsymbol{x})$:

$$\text{NIPV}(\boldsymbol{x}; p_*) = -\mathbb{E}_{\boldsymbol{x}_* \sim p_*}\left[\sigma^2(\boldsymbol{x}_*) \mid \boldsymbol{x}, \mathcal{D}\right]. \tag{3}$$

**Expected Predictive Information Gain** The Expected Predictive Information Gain (EPIG) (Bickford Smith et al., 2023) selects a candidate point $\boldsymbol{x}$ by maximizing the mutual information between its label $y(\boldsymbol{x})$ and the label $y(\boldsymbol{x}^*)$ at a randomly drawn test input $\boldsymbol{x}_* \sim p_*(\boldsymbol{x})$,

$$\text{EPIG}(\boldsymbol{x}; p_*) = \mathbb{E}_{\boldsymbol{x}_* \sim p_*}\left[I\big(y(\boldsymbol{x}); y(\boldsymbol{x}_*) \mid \mathcal{D}\big)\right]. \tag{4}$$

which can equivalently be expressed in terms of an expectation over a difference of entropies. In its general, model-agnostic form, this constitutes of a nested expectation over both test inputs $\boldsymbol{x}_*$ and outcomes and selected locations $y(\boldsymbol{x})$:

$$\text{EPIG}(\boldsymbol{x}; p_*) = \mathbb{E}_{\boldsymbol{x}_* \sim p_*}\big[\text{H}[y(\boldsymbol{x}_*) \mid \mathcal{D}] - \mathbb{E}_{y(\boldsymbol{x})|\mathcal{D}}[\text{H}[y(\boldsymbol{x}_*) \mid \mathcal{D}, \boldsymbol{x}, y(\boldsymbol{x})]]\big]. \tag{5}$$

**EPIG for Gaussian Processes** Standard GP regression with Gaussian likelihoods constitutes a special case for EPIG, as the predictive variance (and therefore the predictive entropy) after conditioning on $(\boldsymbol{x}, y(\boldsymbol{x}))$ is independent of the realized value $y(\boldsymbol{x})$. Thus,

$$\text{H}[y(\boldsymbol{x}_*) \mid \mathcal{D}, \boldsymbol{x}, y(\boldsymbol{x})] = \text{H}[y(\boldsymbol{x}_*) \mid \mathcal{D}, \boldsymbol{x}],$$

and the inner expectation over $y(\boldsymbol{x})$ in (5) disappears. EPIG therefore simplifies to an expected reduction in predictive entropy:

$$\text{EPIG}_{\text{GP}}(\boldsymbol{x}; p_*) = \mathbb{E}_{\boldsymbol{x}_* \sim p_*}[\text{H}[y(\boldsymbol{x}_*) \mid \mathcal{D}] - \text{H}[y(\boldsymbol{x}_*) \mid \mathcal{D}, \boldsymbol{x}]]. \tag{6}$$

We will use this GP-specific form of EPIG throughout the remainder of this paper.

Both NIPV and EPIG promote the selection of data that reduces uncertainty over the test distribution, but without considering the effect on hyperparameter learning. The test distribution encodes how much emphasis is put on different parts of the domain. It can be specified by subject matter experts; in the case of no prior knowledge it is typically the uniform distribution. Throughout the remainder of the paper, we will exclusively consider NIPV with a uniform $p_*$.

## 3   Related Work

Initialization has received surprisingly little attention in the context of BO. Bossek et al. (2020) conducts a study on the effect of various random initial designs on BO performance. Alternatively, minimax or maximin criteria (Johnson et al., 1990) may be used to accomplish evenly distributed designs. Maybe closest to our work is (Zhang et al., 2019), which proposes LHS-Beta, an initial design criterion which alters samples drawn by LHS to achieve pairwise distances between points which matches a Beta distribution. LHS-Beta pursues diverse pairwise distances in the data, in order to best learn the lengthscale of a GP with an isotropic kernel. Müller and Zimmerman (1999); Zimmerman (2006) address the problem of learning parameters of Kriging estimators using a empirical estimates of optimal experimental design criteria, limiting candidates to a fixed grid of points.

Entropy-maximizing (Guestrin et al., 2005; MacKay, 1992, 1995) or variance-minimizing (Park et al., 2024) designs have been explored in active learning for optimal sensor placement (Krause et al., 2006, 2008) and other applications involving GPs, such as geostatistics (Sauer et al., 2023) and contour finding (Cole et al., 2023). Moreover, parameter-related EIG criteria are a bedrock of the broader topic of BED (Bickford Smith et al., 2023; Chaloner and Verdinelli, 1995; Kirsch et al., 2021; Rainforth et al., 2024), which focuses on selecting data that is most informative about model parameters or future predictions. These prediction or model-oriented criteria have yet to see widespread use in BO, particularly for initialization. However, information-theoretic acquisition functions (Hennig and Schuler, 2012; Hernández-Lobato et al., 2014; Hvarfner et al., 2022; Moss et al., 2021; Neiswanger et al., 2021, 2024; Tu et al., 2022; Wang and Jegelka, 2017) address the optimization problem from an information theoretic perspective, albeit not with a primary focus on initialization or model predictive performance.

The problem of actively learning model hyperparameters during BO (post initialization) has previously been investigated by Hvarfner et al. (2023), who propose a combined BO-BAL framework to actively learn the hyperparameters of the GP along with the optimum, and demonstrate that better-calibrated surrogates significantly enhance BO performance. Houlsby et al. (2011) proposes BALD, an active learning acquisition functions for hyperparameters in preference learning in GPs. Riis et al. (2022) proposes a a Query-By-Committee-oriented acquisition function for BAL in GPs. Lastly, Berkenkamp et al. (2019); Ziomek et al. (2024) address BO performance under hyperparameter uncertainty from a theoretical perspective, proving regret bounds when hyperparameters of the objective are unknown.

## 4   Method

We consider the case of large-batch, few-shot BO, where the batch size $q$ is large ($q \geqslant 8$), and only a small number of sequential batches $B$ can be evaluated (often only one for initialization and one for BO, so $B = 2$). With only a handful of batches, there is little opportunity to correct for poor initialization. If the initial design fails to reduce uncertainty in key hyperparameters or leaves large regions of the input space unexplored, both hyperparameter inference and optimization become challenging. This setting is common if evaluations are lengthy but can be effectively parallelized, which is the case for instance in online A/B tests or lab experiments for biology González-Duque et al. (2024); Stanton et al. (2022) or material design Yik et al. (2025).

In practice, the objective(s) often exhibit complex structure such as moderate or high dimensionality $D$ ($D > 5$) and significant uncertainty in the lengthscales $\ell$, noise and signal variance $\sigma_\varepsilon^2$ and $\sigma_f^2$, respectively, and the importance of each input dimension. Standard space-filling designs, such as scrambled Sobol sequences, are widely used for initialization due to their theoretical uniformity properties. However, these methods are agnostic to the underlying model and its uncertainties. As a result, they often fail to uncover critical hyperparameter dependencies (Zhang et al., 2019), leading to poorly calibrated surrogate models and suboptimal acquisition decisions in the subsequent BO phase.

While Sobol and other space-filling sequences are designed to uniformly cover the input space, this property alone does not imply that the resulting model will be desirably helpful for subsequent BO. In fact, space-filling is not synonymous with informativeness: a model trained on a space-filling design may still exhibit high predictive uncertainty in regions most relevant for optimization. For example, random designs increasingly sample near the boundaries of the search space as dimensionality grows Köppen (2000); Swersky (2017), whereas a point located at the center of the search space may be far more informative for optimization, as it can yield a greater reduction in predictive variance throughout the search space.

To this point, a strategy that explicitly reduces predictive uncertainty within the region of interest Bickford Smith et al. (2023)—namely, the search space—not only seems better suited for our goals, but is also better aligned with both theoretical (Berkenkamp et al., 2019; Srinivas et al., 2010) and information gain-based criteria, as well as the practical objective of achieving a more informed, in an informal sense, model post-initialization Garnett (2023). Methods like NIPV (Zhang et al., 2019) which minimize predictive variance over the input space achieve this goal better than Sobol, and yields designs that are spread out but not necessarily informative for model calibration and hyperparameter learning.

On the other hand, methods that focus exclusively on hyperparameter informativeness, such as BALD (Houlsby et al., 2011), tend to cluster queries along specific axes in order to resolve length-scales, thereby sacrificing coverage and potentially overlooking important regions of the search space. When additional hyperparameters are present, such as a learnable noise variance, BALD may even select duplicate queries at the same location to better estimate said noise variance. While such behavior is not inherently problematic, it can be undesirable in the few-shot setting, where initialization must address hyperparameter learning and space-filling simultaneously.

To overcome these limitations, we introduce HIPE, a method that explicitly balances predictive uncertainty reduction with hyperparameter-awareness. By jointly considering coverage and informativeness, HIPE produces initial designs that effectively reduce both predictive and hyperparameter uncertainty, ensuring robust model calibration and improved downstream optimization.

The interplay between all these initialization strategies is illustrated in Figure 1, which visualizes the acquisition surfaces for a GP model in two dimensions under lengthscale uncertainty. Each subplot highlights the distinct behavior of a different method. Sobol emphasizes uniform coverage of the input space, but its designs may not achieve the desired reduction in predictive uncertainty, especially given the inherent randomness and lack of model awareness. BALD, in contrast, focuses on reducing lengthscale uncertainty by selecting axis-aligned queries, which can lead to clustering along specific directions and a loss of broader coverage. NIPV aims to minimize average predictive variance across the input space, resulting in well-spread points, but it can neglect hyperparameter informativeness since its queries are not tailored to resolve model parameter uncertainty. Our proposed method, HIPE, strikes a balance between predictive uncertainty reduction and hyperparameter-awareness: it selects points that are both well-distributed and aligned with the axes of uncertainty, thereby achieving low predictive and hyperparameter uncertainty simultaneously.

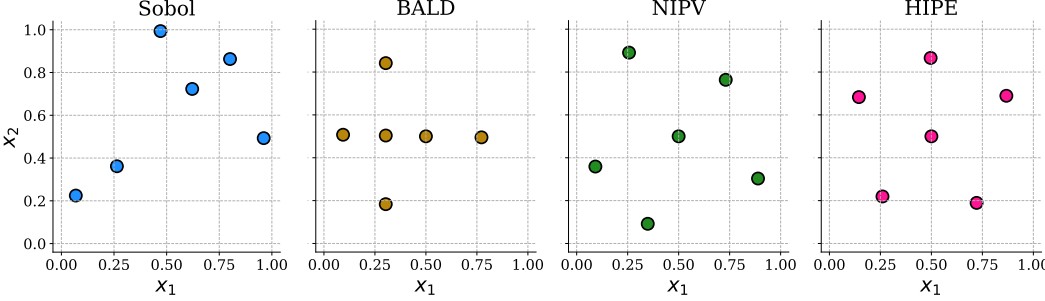

**Figure 1:** Visual comparison of initialization strategies for BO with a GP in two dimensions under lengthscale uncertainty. Each plot shows the acquisition surface used to guide batch selection. Sobol emphasizes space-filling, but may not accomplish this to the desired degree. BALD focuses on reducing lengthscale uncertainty via axis-aligned queries. NIPV spreads points to minimize average predictive variance over the input space. Lastly, HIPE balances space-filling and hyperparameter-awareness, choosing spread-out points while preserving axis-alignment between queries.

## 4.1 Hyperparameter-Informed Predictive Exploration

We approach the problem of initialization through the lens of optimization, by simultaneously maximizing the coverage of a region of interest and the information acquired about model-level uncertainty. A natural way to achieve this is to optimize a criterion that combines EPIG and BALD:

$$\text{HIPE}_\beta(\boldsymbol{X}) := \underbrace{-\mathbb{E}_{\boldsymbol{\theta}, y(\boldsymbol{X})}\left[\mathbb{E}_{\boldsymbol{x}_*}\left[\text{H}\left[y(\boldsymbol{x}_*) \mid \boldsymbol{\theta}, y(\boldsymbol{X})\right]\right]\right]}_{\text{EPIG objective}} + \beta \underbrace{\left(\text{H}[y(\boldsymbol{X})] - \mathbb{E}_{\boldsymbol{\theta}}\left[\text{H}[y(\boldsymbol{X}) \mid \boldsymbol{\theta}]\right]\right)}_{\text{BALD objective}} \quad (7)$$

where $\beta > 0$ is a scalar weighting parameter. For large $\beta$, this objective favors hyperparameter learning, while for small $\beta$ it favors space-filling designs.

It turns out that for the choice of $\beta = 1$, the maximizer of Eq. (9) is exactly the maximizer of the joint information gain over test function values and model hyperparameters (for proof see Appendix B):

**Proposition 1** (Equivalence of HIPE$_{\beta=1}$ to Joint Information Gain). *The* HIPE$_\beta$ *acquisition function with* $\beta = 1$ *is equivalent to maximizing the expected joint information gain over test function values* $y(\boldsymbol{x}*)$ *and model hyperparameters* $\boldsymbol{\theta}$ *acquired by a candidate batch* $\boldsymbol{X}$. *Formally,*

$$\underset{\boldsymbol{X} \in \mathbb{R}^{q \times D}}{\arg\max} \ \text{HIPE}_{\beta=1}(\boldsymbol{X}; p_*) = \underset{\boldsymbol{X} \in \mathbb{R}^{q \times D}}{\arg\max} \ \mathbb{E}_{\boldsymbol{x} \sim p_*}\left[\text{EIG}\left(y(\boldsymbol{x}_*), \boldsymbol{\theta}; \boldsymbol{X}\right)\right]. \quad (8)$$

While Proposition 1 provides an intuitive connection between HIPE, EPIG, and BALD, the constituent quantities often have vastly different scales. Therefore the choice of $\beta = 1$ will generally not result in optimal performance since the optimization will inadvertently focus on the larger of the two terms.

A key observation is that not all hyperparameter information gain amounts to information gained on the test set – this depends on multiple aspects, including downstream test distribution and hyperparameterization. To what extent the reduction in hyperparameter entropy *manifests in a reduction in test set entropy* can be quantified through the mutual information between the hyperparameters $\boldsymbol{\theta}$ and the test points $y(\boldsymbol{x}_*)$, $\boldsymbol{x}_* \sim p_*(\boldsymbol{x})$:

$$\beta = \text{EIG}(y(\boldsymbol{x}_*); \boldsymbol{\theta}|\mathcal{D}) = \mathbb{E}_{\boldsymbol{x}_*}\left[\text{H}[y(\boldsymbol{x}_*)|\mathcal{D}] - \mathbb{E}_{\boldsymbol{\theta}}[\text{H}[y(\boldsymbol{x}_*)|\boldsymbol{\theta}, \mathcal{D}]]\right]. \quad (9)$$

Intuitively, Eq. (9) quantifies how well the knowledge of the hyperparameters, in expectation, informs us about the values of $y(\boldsymbol{x}_*)$. Importantly, $\text{EIG}(y(\boldsymbol{x}_*); \boldsymbol{\theta}|\mathcal{D})$ does not depend on the candidate set $\boldsymbol{X}$ and can thus be pre-computed.

Setting $\beta = \text{EIG}(y(\boldsymbol{x}_*); \boldsymbol{\theta}|\mathcal{D})$ balances the two competing objectives in Eq. (7) according to their effect on downstream predictive uncertainty, without introducing any additional hyperparameters. We refer to the resulting acquisition function as Hyperparameter-Informed Predictive Exploration (HIPE):

$$\text{HIPE}(\boldsymbol{X}) := -\mathbb{E}_{y(\boldsymbol{X})}\left[\mathbb{E}_{\boldsymbol{x}_*}[\text{H}[y(\boldsymbol{x}_*)|\boldsymbol{\theta}, y(\boldsymbol{X})]]\right] + \text{EIG}(y(\boldsymbol{x}_*); \boldsymbol{\theta}) \, \mathbb{E}_{\boldsymbol{\theta}}[\text{H}[y(\boldsymbol{X})|\boldsymbol{\theta}] - \text{H}[y(\boldsymbol{X})]] \quad (10)$$

Notably, optimizing HIPE does not require having observed *any* data – for this problem to be well-posed only requires test distribution, model structure, and model parameter hyperpriors.

## 4.2 A Parallel Monte Carlo Implementation of HIPE

Following the Monte Carlo approach used in modern acquisition functions (Balandat et al., 2020; Wilson et al., 2020), we implement a parallel version of HIPE that enables joint optimization. For a candidate batch $\boldsymbol{X} = \{\boldsymbol{x}_1, \dots, \boldsymbol{x}_q\}$, we estimate the acquisition objective using $M$ MC samples over the hyperparameters, $T$ test locations of the EPIG objective, and $N$ samples from the predictive posterior for the BALD objective:

$$\alpha_{\text{BALD}}(\boldsymbol{X}; \boldsymbol{\theta}) = -\log p\left(\mathbb{E}_{\boldsymbol{\theta}}[y(\boldsymbol{X})|\boldsymbol{\theta}]\right) + \mathbb{E}_{\boldsymbol{\theta}}\left[\log p(y(\boldsymbol{X}) \mid \boldsymbol{\theta})\right] \quad (11a)$$

$$\approx -\frac{1}{N} \sum_{n=1}^{N}\left[\log\left(\frac{1}{M}\sum_{m=1}^{M} p\left(Y^{(n)} \mid \theta^{(m)}\right)\right) + \frac{1}{M}\sum_{m=1}^{M}\log p\left(Y^{(n)} \mid \theta^{(m)}\right)\right], \quad (11b)$$

where $\boldsymbol{\theta}^{(m)} \sim p(\boldsymbol{\theta} \mid \mathcal{D})$ are i.i.d samples from the belief over hyperparameters, and $Y^{(n)} \sim y(\boldsymbol{X})$ are i.i.d, $q$-dimensional (joint) samples from the predictive posterior. Thus, the BALD objective amounts

to repeated evaluation the $q$-dimensional multivariate normal posterior $y(\boldsymbol{X})$ for each candidate $\boldsymbol{X}$, and estimating the posterior entropy from it. Secondly, we estimate the EPIG objective analogously to Balandat et al. (2020) as

$$\alpha_{\text{EPIG}}(\boldsymbol{X}; p_*, \boldsymbol{\theta}) \approx \frac{1}{M}\frac{1}{T}\sum_{m=1}^{M}\sum_{t=1}^{T}\left[C - \mathrm{H}[y(\boldsymbol{x}_*^{(t)})|\boldsymbol{X}, \boldsymbol{\theta}^{(m)}, \mathcal{D}]\right], \qquad (12)$$

where $C = \mathrm{H}[y(\boldsymbol{x}_*^{(t)})|\mathcal{D}]$ is constant w.r.t. $\boldsymbol{X}$ and does not need to be computed. Since $y(\boldsymbol{x}_*^{(t)})|\boldsymbol{X}, \boldsymbol{\theta}^{(m)}, \mathcal{D}$ in Eq. (12) is the GP posterior predictive at $\boldsymbol{x}_*$, it is a Gaussian random variable and its entropy can be computed in closed form without observing or simulating the outcome at $\boldsymbol{x}_*$. Thus, Eq. (12) can be evaluated in closed form for each of the $M$ GPs' $T$ test points after conditioning on the set $\boldsymbol{X}$.

With these two MC estimators, the subsequent optimization can be carried out jointly for the entire $q$-batch in a $qD$-dimensional space, as the entropy reduction is computed with regard to the entire batch of candidates as opposed to a singular one. Using Sample Average Approximation (Balandat et al., 2020), the HIPE objective is deterministic and auto-differentiable.

One downside of this formulation is that the nested MC estimator imposes substantial computational runtime, making HIPE less suited for high-throughput applications (Daulton et al., 2022; Eriksson et al., 2019; Maus et al., 2022). However, in those applications the quality of the initialization batch is generally much less crucial, so this is not a limitation in practice.

## 5  Results

We evaluate HIPE across two main types of tasks: Active Learning (AL) and Bayesian Optimization (BO). We consider various synthetic and real-world problems and a number of different baselines. In both settings, we maintain large batch sizes and few batches.

**Setup**  For AL, we simply run a number of batches with HIPE and the other baselines with the goal of achieving the best model fit. For BO, we consider the "two-shot" setting in which we first have to select an initialization batch and then can perform a single iteration of (batch) BO using qLogNEI (Ament et al., 2023a) as the acquisition function. We benchmark against the conventional initializations Sobol and Random Search, as well as BALD, NIPV and LHS-Beta (Zhang et al., 2019). On all tasks, we utilize a fully Bayesian GP (Eriksson and Jankowiak, 2021; Snoek et al., 2012) using MCMC with NUTS (Hoffman and Gelman, 2014) in Pyro (Bingham et al., 2018). We implement HIPE and all baselines in BoTorch (Balandat et al., 2020). For all experiments, the hyperparameter set $\boldsymbol{\theta}$ consists of lengthscales $\boldsymbol{\ell}$ for each dimension with a $D$-scaled prior (Hvarfner et al., 2024), a constant mean $c$, and an inferred noise variance $\sigma_\varepsilon^2$ unless otherwise specified. All baselines, including random algorithms, are given the center of the search space as part of their initial design, as both HIPE and NIPV select the center of the search space by design under a uniform $p_*$. Complete details on the experimental setup, including code pointers and benchmarks can be found in Appendix A.

**Evaluation Criteria**  We measure model fit quality with Root Mean Square Error (RMSE) of the mean prediction and Negative Log-Likelihood (NLL) against a large number of (ground truth) test point sampled uniformly from the domain. In the "two-shot" optimization setting, we are interested in how the initialization affects the quality of the GP surrogate after both the first (initialization) and second (BO) batch. We compute relative rankings, the performance of each algorithm compared to its competitors, for each seed of each function, and average across the task type. As such, the relative rankings aggregate inter-algorithm performance across rows for Figs 2- 4.

We also study how these model quality improvements translate to better optimization performance. To this end, we consider the *out-of-sample inference performance* (Hernández-Lobato et al., 2014; Hvarfner et al., 2022), that is, the performance of the point $\boldsymbol{x}' = \arg\max \mu(\boldsymbol{x} \mid \mathcal{D})$ selected as the maximizer of the posterior mean of the surrogate model fit on the data available in each batch. We choose this metric since using observed points directly performs very poorly in noisy settings, and only considering in-sample points is rather limiting in the few-shot setting.

## 5.1 Batch Active Learning on GP Surrogates and Synthetic Functions

We first evaluate the ability of HIPE to learn accurate surrogate models through batch active learning on noisy synthetic test functions and surrogate LCBench (Zimmer et al., 2021) tasks. The LCBench tasks are derived from complete neural network training runs on various OpenML (Vanschoren et al., 2014) datasets, with 7D GP surrogate models fitted as described in Appendix A. Additionally, we evaluate performance on the Hartmann 6D function and a high-dimensional Hartmann 6D (12D) variant, where dummy input dimensions are added following standard practice in high-dimensional BO (Eriksson and Jankowiak, 2021). These dummy dimensions introduce an additional challenge, as effectively identifying and ignoring irrelevant features is critical for accurate predictions. All evaluations are subject to substantial observation noise, detailed in Appendix A.

We run each algorithm for 4 batches of size $q = 16$ and measure the NLL and RMSE after each batch, displaying mean and one standard error on all tasks. In Fig. 2 we see that, across all tasks, HIPE is the only method that consistently ranks in the top two for both RMSE and NLL, demonstrating that the models it produces are both accurate and well-calibrated. On NLL, HIPE performs comparably to BALD, which targets hyperparameter learning and thus excels at model calibration. Similarly, HIPE is competitive with NIPV on RMSE, a metric for which NIPV is particularly well-suited (Gramacy and Lee, 2009).

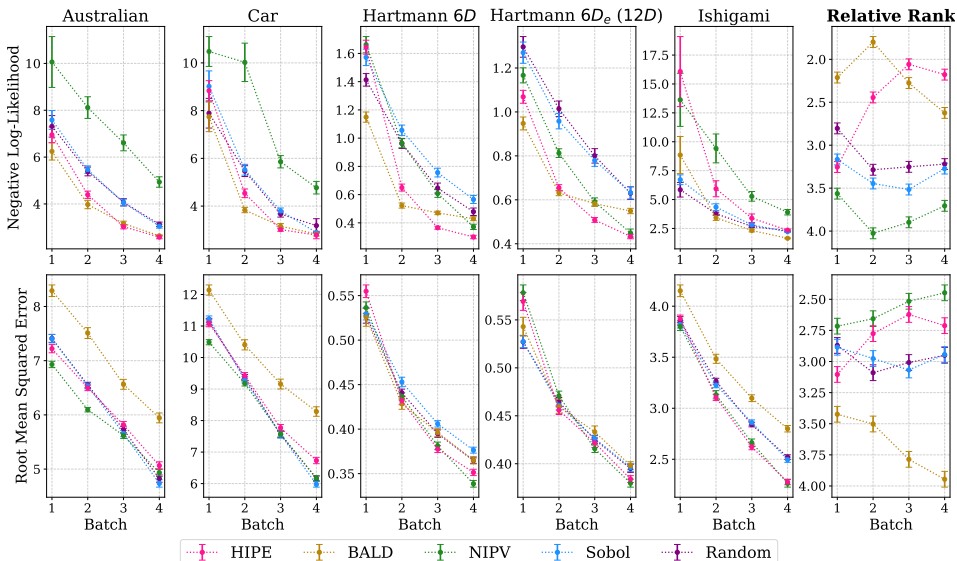

**Figure 2:** Model accuracy results in the batch active learning setting. We report RMSE across various synthetic and LCBench surrogate tasks over 4 batches of $q = 16$ evaluations across 100 seeds. HIPE consistently ranks in the top two in relative rankings on both metrics, achieving a strong balance between hyperparameter learning and predictive accuracy. BALD performs competitively on marginal log-likelihood (MLL) but underperforms on RMSE due to poor space-filling, while NIPV excels at reducing RMSE but struggles with calibration. On aggregate, random initialization methods lag significantly behind across all benchmarks.

## 5.2 Noisy Synthetic Test Functions

Next, we evaluate HIPE on synthetic benchmark functions in the two-shot Bayesian optimization setting, using $B = 2$ batches and a batch size of $q = 24$. We consider three standard test functions—Ackley (4D), Hartmann (4D), and Hartmann (6D)—as well as two higher-dimensional variants of the Hartmann function. Observation noise is added to all tasks ($\sigma_\varepsilon = 2$ for Ackley and $\sigma_\varepsilon = 0.5$ for Hartmann), further increasing task difficulty. In Fig. 3 shows that across all benchmarks, HIPE consistently achieves the best or second-best performance, followed by NIPV. Random and space-filling initialization methods (LHS-Beta, Random, and Sobol) perform noticeably worse across all settings. In App. C.1, we show the runtime of HIPE and various other methods on Hartmann (6D). The extra cost of HIPE acquisition is a minority of iteration runtime for a fully Bayesian model.

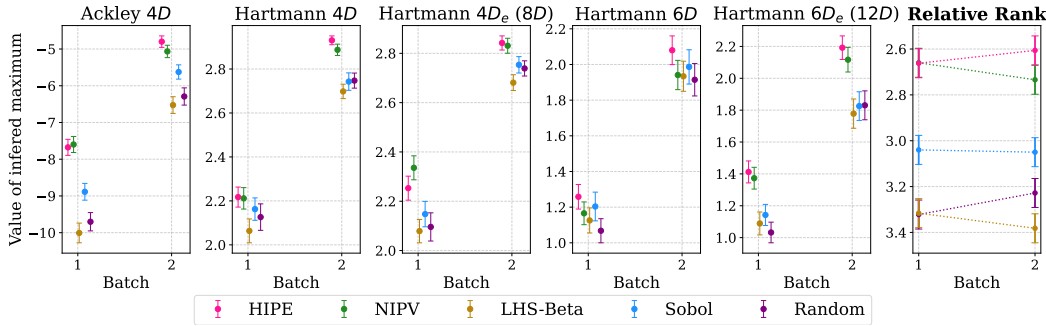

**Figure 3:** Out-of-sample inference optimization performance on noisy synthetic benchmark functions under the two-shot setting ($B = 2$, $q = 24$) across 100 seeds per benchmark. HIPE outperforms or matches the best-performing method across all benchmarks, including on the high-dimensional Hartmann variants with added dummy variables. NIPV performs well on most tasks, but its performance degrades on tasks where hyperparameter identification is critical. Random initialization strategies perform the worst throughout.

## 5.3 LCBench HPO Tasks

We evaluate five additional tasks from LCBench in the two-shot optimization setting: Fashion-MNIST, MiniBooNE, Car, Higgs, and Segment, using fixed, minimal observation noise. Fig. 4 displays that HIPE substantially outperforms the competition on Fashion-MNIST, Mini-BooNE and Higgs, and performs competitively on the remaining Segment and Car. Overall, HIPE consistently delivers the highest relative rank, outperforming competing algorithms by a substantial margin. In App. 7, we demonstrate the performance of the same initialization schemes on $q = 8$, $B = 5$. In this setting, HIPE remains the top-performing method, and the overall ranking of algorithms remains intact.

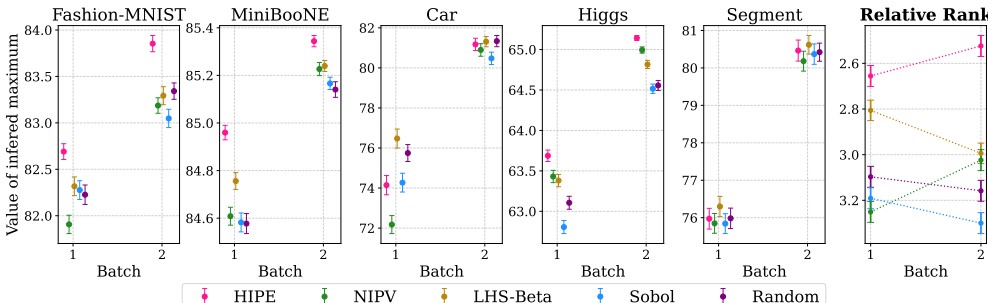

**Figure 4:** Two-shot optimization results on five hyperparameter optimization tasks from LCBench across 200 seeds for each task. HIPE achieves the highest final performance on four of the five tasks and remains competitive on the remaining one (Segment). Relative rankings across the five problems show that HIPE consistently outperforms other methods. This confirms the practical relevance of our approach for real-world HPO scenarios, where both accurate predictive modeling and effective exploration are crucial.

## 5.4 High-Dimensional SVM Tasks

Finally, we evaluate HIPE and baseline methods on challenging high-dimensional SVM hyperparameter optimization tasks with $D = 20$ and $D = 40$ input dimensions, considered in similar variants by Ament et al. (2023a); Eriksson and Jankowiak (2021); Hvarfner et al. (2024); Papenmeier et al. (2023). For both tasks, only the last two dimensions—corresponding to the SVM's global regularization parameters—significantly influence the objective, while the remaining dimensions, corresponding to feature-specific lengthscales, are of lesser importance. Effectively identifying and prioritizing these relevant dimensions is critical for successful optimization. We again consider the two-shot setting, using a larger batch size of $q = 32$ due to the higher dimensionality of the problems.

The left panel of Fig. 5 reports the out-of sample inference performance after each batch. On the 20D task, HIPE achieves competitive, mid-range performance relative to the evaluated methods. On the more challenging 40D task, HIPE obtains the highest performance, albeit by a narrow margin over the next-best alternatives. The limited budget relative to the dimensionality presents a

significant challenge, as the ability to accurately learn the model hyperparameters diminishes with increasing dimensionality. Despite this, we observe in the right panel of Fig. 5 that HIPE identifies the important hyperparameters remarkably well after initialization on the 40D task—assigning the last two dimensions lengthscales that are, on average, nearly half an order of magnitude smaller than those inferred under a Sobol initialization. In Fig.10, we display the inferred hyperparameter values for NIPV and Random as well. While these methods infer lower lengthscales for the last two dimensions than Sobol, they do not manage to infer as low values as HIPE.

Finally, we note that the best solutions to the SVM problem were almost exclusively located near the boundaries of the search space—particularly in the last two dimensions, which neither NIPV nor HIPE naturally explore during initialization.

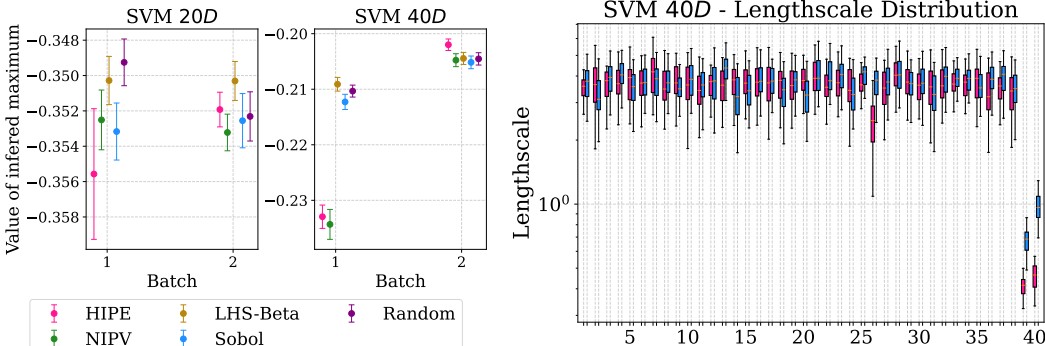

**Figure 5: Results on 20D and 40D SVM hyperparameter optimization tasks. Left:** Objective value of inferred maximum after each batch. On 20D, HIPE achieves average performance, climbing to a mid-tier result in the second batch after focusing on hyperparameter learning in the first batch. HIPE obtains a large standard error in the first batch, as two repetitions poorly infers the maximizer and suggests ill-performing points as a result. On 40D, it outperforms all baselines, demonstrating strong robustness in higher dimensions and the ability to recover after a less successful first batch. Notably, the performance of Random decreases between batches on the 20D task, demonstrating the difficulty of accurately inferring the optimum. **Right:** Log-mean estimated lengthscale hyperparameters *after initialization* on the 40D task. HIPE identifies the last two dimensions–corresponding to the SVM's global regularization parameters – much more effectively than Sobol, assigning substantially smaller lengthscales to the relevant inputs.

## 6 Discussion

**Contributions** We introduced HIPE, a principled, hyperparameter-free information-theoretic method for initializing Bayesian Optimization and Bayesian Active Learning algorithms. its HIPE yields initial designs that balance coverage of (relevant) areas of the domain with the ability to effectively learn model hyperparameters. HIPE is especially useful in the few-shot, large-batch setting, where it achieves superior surrogate model quality compared to various initialization baselines.

**Limitations** HIPE can become computationally expensive, especially for large batch sizes $q$ in higher dimensions $D$, though in many applications this cost is still insignificant compared to the time and resources required to evaluate the underlying black-box function. Finally, the current paper focuses on GP surrogates, and while our main insights and the general approach apply also to other surrogate types, our implementation does not translate directly. Thus, the computation of the acquisition function would have to be re-derived for other types of Bayesian models.

**Future work** Here we studied the "cold start" problem of initializing Bayesian Optimization and Bayesian Active Learning from scratch. In practice, we may have access to data from related (but not necessarily identical) problems. This motivates an extension of HIPE to the transfer learning setting, e.g., by means of using a multi-task GP surrogate. Additionally, incorporating prior knowledge of domain experts in a principled fashion is of high practical relevance, and can readily be utilized in the form of a non-uniform $p_*$. Finally, we are also interested in studying the multi-objective setting in which different surrogates of potentially different form with different hyperparameters and priors model different objectives but share observations at the same input locations.

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

# A    Experimental Setup

We describe the full experimental setup used in the paper: the design of the Bayesian optimization and active learning loops, the benchmarks used, the compute as well as the licenses of all software and datasets. The code to run the experiments in the paper, including all the benchmarks and plotting to reproduce our results, is available at `https://github.com/hipeneurips/HIPE`.

An easy-to-run, tested version of the algorithm and accompanying notebook is available open-source in BoTorch. The HIPE acquisition is found at `https://github.com/meta-pytorch/botorch/blob/main/botorch_community/acquisition/bayesian_active_learning.py`. The notebook, which demonstrates how to run the algorithm should be run in its indented setting and the expected results, can be found at `https://github.com/meta-pytorch/botorch/blob/main/notebooks_community/hyperparameter_informed_predictive_exploration`.

## A.1    Bayesian Optimization Loop

All experiments were conducted using a standardized pipeline based on BoTorch (Balandat et al., 2020) and GPyTorch (Gardner et al., 2018). We use fully Bayesian Gaussian Process (GP) models, with hyperparameter inference performed via the No-U-Turn Sampler (NUTS )(Hoffman and Gelman, 2014) as implemented in Pyro (Bingham et al., 2018). Unless otherwise specified, we draw 192 burn-in samples followed by 288 hyperparameter samples, retaining every 24$^{\text{th}}$ sample for evaluation.

Our GP prior is adapted from Hvarfner et al. (2024) to better suit a fully Bayesian setting. Specifically, we set $\mu_0 = -0.75$ and $\sigma = 0.75$, resulting in $\ell_d \sim \mathcal{LN}(0.75 + \log(D)/2, 0.75)$. The noise standard deviation is modeled as $\sigma_\varepsilon \sim \mathcal{LN}(-5.5, 0.75)$, and the constant mean parameter follows $c \sim \mathcal{N}(0, 0.25)$. These modifications ensure that the means of the priors for $\ell_d$ and $\sigma_\varepsilon^2$ approximately match the modes of the corresponding parameters in the prior proposed by Hvarfner et al. (2024), producing similar hyperparameter values in practice.

Acquisition functions are optimized jointly over the batch using multi-start L-BFGS-B optimization with 4 random restarts and 384 initial samples drawn from a scrambled Sobol sequence. For the optimization of `LogEI`, we additionally sample 384 points from a Gaussian distribution centered at the current incumbent to improve local search performance.

For our proposed HIPE method and relevant baselines (BALD and NIPV), Monte Carlo estimators use $M = 12$ hyperparameter samples, $T = 1024$ test points drawn uniformly from the search space, and $N = 128$ predictive posterior samples.

## A.2    Benchmarks

We use three types of benchmarks: synthetic optimization test functions, surrogate-based hyperparameter optimization tasks from LCBench (Zimmer et al., 2021), and high-dimensional SVM hyperparameter optimization problems. Synthetic functions are standard benchmarks for evaluating active learning and Bayesian optimization under controlled noise and dimensionality. LCBench tasks are GP surrogate models trained on 2,000 evaluations of multi-layer perceptrons (MLPs) on real-world datasets, using a Matern 3/2 kernel; all surrogates are included as part of our code release. The SVM benchmarks follow the setup in Ament et al. (2023a), based on the problem originally introduced in Eriksson and Jankowiak (2021). For these tasks, a Support Vector Regressor (Drucker et al., 1996) is fit to the CTSlice dataset, using a fixed subset of 5,000 data points, with 20% reserved for validation. To vary the dimensionality, an XGBoost model is fit to the original 388-dimensional dataset, and the most important features are retained according to XGBoost's default feature selection criterion. The objective is to minimize the validation RMSE. In synthetic functions, additive Gaussian noise is introduced directly to the function evaluations (see Table 1 for details). The LCBench benchmarks rely on pre-trained surrogate models, where the posterior mean is evaluated, and in cases where $\sigma_\varepsilon$ is non-zero, additional noise is added to the evaluation of the posterior mean.

## A.3    Licenses

The following software packages, libraries and datasets were used in our experiments and for presenting the results in the paper:

**Table 1:** Noise levels for all Active Learning (AL) and Bayesian Optimization (BO) benchmarks.

| Category | Benchmark | Task Type | Dimensionality | $\sigma_\varepsilon$ |
|---|---|---|---|---|
| Synthetic | Ackley (4D) | BO | 4 | 2.0 |
| | Hartmann (6D) | BO / AL | 6 | 0.5 |
| | Hartmann (6D) | BO / AL | 12 | 0.5 |
| | Hartmann (4D) | BO | 4 | 0.5 |
| | Hartmann (4D) | BO | 8 | 0.5 |
| LCBench | Car | AL | 7 | 2.5 |
| | Australian | AL | 7 | 2.5 |
| | Fashion-MNIST | BO | 7 | 0.0 |
| | MiniBooNE | BO | 7 | 0.0 |
| | Car | BO | 7 | 0.0 |
| | Higgs | BO | 7 | 0.0 |
| | Segment | BO | 7 | 0.0 |
| SVM | Feature-reduced SVM | BO | 20 | 0.0 |
| | Feature-reduced SVM | BO | 40 | 0.0 |

- **GPyTorch**, **BoTorch**, **Hydra**: MIT License
- **PyTorch**, **NumPy**, **SciPy**, **Pandas**: BSD Licenses
- **Matplotlib**, **Seaborn**: PSF/BSD Licenses
- **LCBench**: Apache License

## A.4 Compute Resources

All experiments were conducted using an NVIDIA A40 GPU cluster. The compute usage to run all experiments in the main paper amounts to approximately 1000 GPU hours, and an additional 500 GPU hours to produce all the results provided in Appendix C.

## B Derivation of HIPE as Joint Information Gain

Recall Proposition 1 from section 4.1.

**Proposition 1** (Equivalence of HIPE$_{\beta=1}$ to Joint Information Gain)**.** *The* HIPE$_\beta$ *acquisition function with $\beta = 1$ is equivalent to maximizing the expected joint information gain over test function values $y(x_*)$ and model hyperparameters $\boldsymbol{\theta}$ acquired by a candidate batch $\boldsymbol{X}$. Formally,*

$$\underset{\boldsymbol{X} \in \mathbb{R}^{q \times D}}{\arg\max} \ \text{HIPE}_{\beta=1}(\boldsymbol{X}; p_*) = \underset{\boldsymbol{X} \in \mathbb{R}^{q \times D}}{\arg\max} \ \mathbb{E}_{\boldsymbol{x} \sim p_*} \left[ \text{EIG}\left( y(\boldsymbol{x}_*), \boldsymbol{\theta}; \boldsymbol{X} \right) \right]. \tag{8}$$

*Proof of Proposition 1.* For $\beta = 1$, we have

$$\text{HIPE}_1(\boldsymbol{X}) = -\mathbb{E}_{\boldsymbol{\theta}, y(\boldsymbol{X})} \left[ \mathbb{E}_{\boldsymbol{x}_*} \left[ \text{H}\left[ y(\boldsymbol{x}_*) \mid \boldsymbol{\theta}, y(\boldsymbol{X}) \right] \right] \right] + \left( \text{H}[y(\boldsymbol{X})] - \mathbb{E}_{\boldsymbol{\theta}} \left[ \text{H}[y(\boldsymbol{X}) \mid \boldsymbol{\theta}] \right] \right)$$

Therefore, with $\boldsymbol{X}^* := \arg\max_{\boldsymbol{X} \in \mathbb{R}^{q \times D}} \text{HIPE}_1(\boldsymbol{X})$,

$$\boldsymbol{X}^* = \underset{\boldsymbol{X} \in \mathbb{R}^{q \times D}}{\arg\max} -\mathbb{E}_{\boldsymbol{\theta}, y(\boldsymbol{X})} \left[ \mathbb{E}_{\boldsymbol{x}_*} \left[ \text{H}\left[ y(\boldsymbol{x}_*) \mid \boldsymbol{\theta}, y(\boldsymbol{X}) \right] \right] \right] + \left( \text{H}[y(\boldsymbol{X})] - \mathbb{E}_{\boldsymbol{\theta}} \left[ \text{H}[y(\boldsymbol{X}) \mid \boldsymbol{\theta}] \right] \right) \tag{13a}$$

$$= \underset{\boldsymbol{X} \in \mathbb{R}^{q \times D}}{\arg\max} \ -\mathbb{E}_{\boldsymbol{\theta}, y(\boldsymbol{X})} \left[ \mathbb{E}_{\boldsymbol{x}_*} [\text{H}[y(\boldsymbol{x}_*) | \boldsymbol{\theta}, y(\boldsymbol{X})]] \right] - \mathbb{E}_{\boldsymbol{\theta}} [\text{H}[\boldsymbol{\theta} | y(\boldsymbol{X})]] \tag{13b}$$

$$= \underset{\boldsymbol{X} \in \mathbb{R}^{q \times D}}{\arg\max} \ \mathbb{E}_{\boldsymbol{x}_*} \left[ \text{H}[y(\boldsymbol{x}_*), \boldsymbol{\theta}] \right] - \mathbb{E}_{y(\boldsymbol{X})} [\text{H}[y(\boldsymbol{x}_*), \boldsymbol{\theta} | y(\boldsymbol{X})]] \quad \text{(by def)} \tag{13c}$$

$$= \underset{\boldsymbol{X} \in \mathbb{R}^{q \times D}}{\arg\max} \ \mathbb{E}_{\boldsymbol{x}_*} [\text{EIG}(y(\boldsymbol{x}_*), \boldsymbol{\theta}; \boldsymbol{X})] \tag{13d}$$

where the equalities follow from the Bayes' rule of conditional entropy and the fact that the $\arg\max$ is independent of quantities which do not involve $\boldsymbol{X}$. $\qquad\square$

# C   Additional Experiments

We present supplementary experiments to further analyze the behavior of the evaluated methods under varying experimental conditions. These results provide additional insights into the robustness of the methods with respect to batch size during initialization and active learning, as well as their performance trade-offs across different evaluation metrics.

## C.1   Runtime Analysis

We compare acquisition function optimization runtime of HIPE and NIPV, and put that in relation to the runtime of the other components of a BO loop that employs fully Bayesian GP modeling. In Table 2 we demonstrate that while HIPE is substantially slower to optimize than NIPV, its total runtime still amounts to only a minor share of the total runtime of the overall BO run.

| Runtime Component (sec) | HIPE | NIPV | Random |
|---|---|---|---|
| Model fit (initial) | $21.91 \pm 0.15$ | $21.91 \pm 0.15$ | N/A |
| Acquisition Opt. | $19.61 \pm 1.12$ | $8.62 \pm 0.37$ | — |
| Model fit (BO loop) | $181.79 \pm 7.60$ | $189.90 \pm 5.17$ | $177.33 \pm 7.69$ |
| qLogNEI Optimization | $4.97 \pm 0.18$ | $4.44 \pm 0.05$ | $4.35 \pm 0.06$ |

**Table 2:** Total runtime comparison (in seconds) for HIPE, NIPV, and Random across different components over the course of a 4-batch, $q = 8$ optimization run. HIPE is more time-consuming to optimize compared to NIPV. While NIPV takes approximately half as long to optimize, the runtime to optimize HIPE is small relative to the time required to fit the fully Bayesian GP with NUTS.

## C.2   Impact of Batch Size on Predictive Performance

We investigate how the initialization batch size $q$ affects predictive quality and inference performance on the LCBench tasks introduced in Section 5.3. We vary $q$ from 6 to 24 in increments of 2 up to $q = 16$, then in steps of 4. For each batch size, we evaluate models trained on the initialization batch only — before any active learning iterations — under a fixed-noise setting. This isolates the impact of the initial design and avoids the confounding effects of subsequent data acquisition.

Fig. 6 summarizes the results across three key metrics: test-set NLL, test-set RMSE, and out-of-sample inference performance. HIPE consistently achieves the lowest NLL across most benchmarks and batch sizes, particularly in the $q \in [12, 16]$ range, demonstrating strong calibration and hyperparameter learning. In contrast, RMSE results show that HIPE often underperforms relative to simpler baselines like Sobol and Random, suggesting a trade-off between predictive calibration and pointwise accuracy. Inference performance shows HIPE leading for smaller batch sizes ($q < 16$), although performance plateaus or declines at higher $q$, indicating that additional samples are not necessarily helpful for inference when uncertainty in relevant parameters remains high. This occurs because, as batch size increases for our experiments, the inferred $\arg\max$ has an increased propensity to be located at a boundary. As these boundary points may not be well-performing, inference performance occasionally stagnates or decreases with increasing batch size. Notably, as HIPE's initialization is generally more centered than competing methods', it obtains higher in-sample values across most batch sizes.

Unlike the active learning setup, these tasks are noiseless and assume fixed, known noise levels during training. This removes uncertainty in the noise model and creates a distinct evaluation setting focused solely on input selection and hyperparameter inference.

## C.3   Bayesian Optimization with Different Batch Sizes

We further investigate the impact of batch size on Bayesian optimization performance by evaluating all methods on the LCBench tasks with a reduced batch size of $q = 8$. As shown in Fig. 6, HIPE maintains its position as the best-performing method overall, achieving the lowest inference regret across the majority of tasks. Notably, HIPE benefits more than competing methods from identifying high-valued points during the initialization phase, which can be attributed to HIPE biasing towards

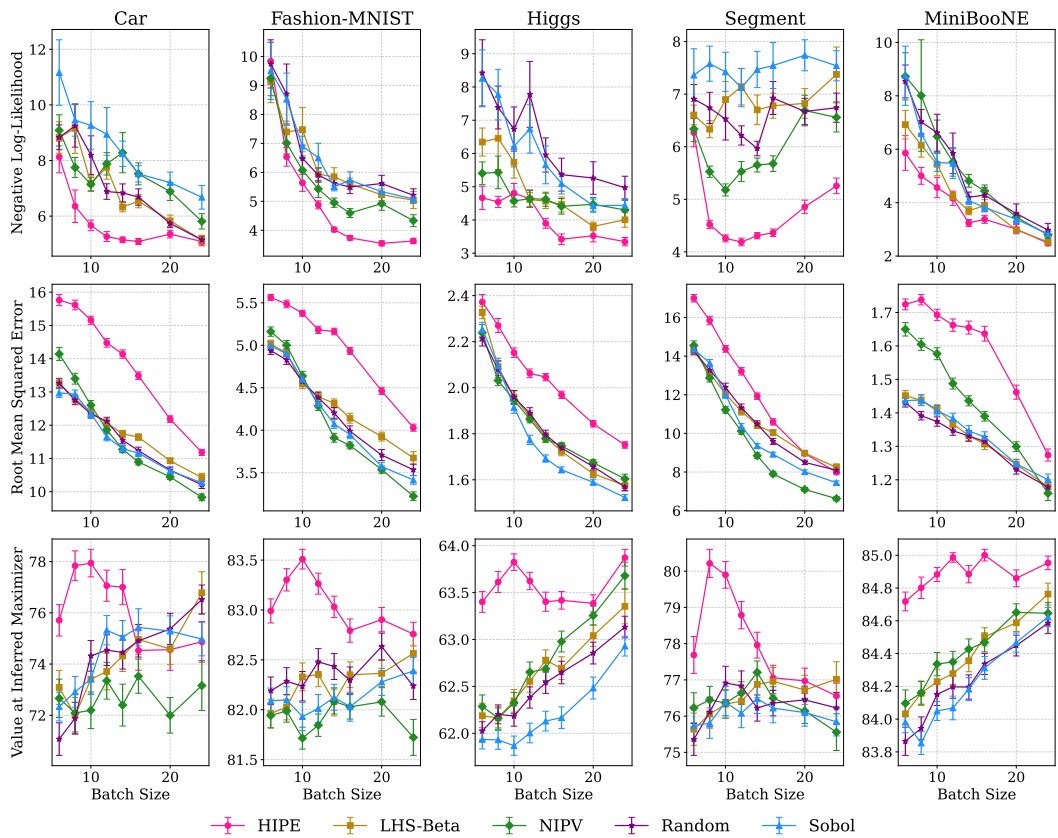

**Figure 6:** Effect of initialization batch size $q$ on predictive quality and inference across LCBench tasks. Each row shows performance on one of three metrics after the initialization batch: (top) NLL, (middle) RMSE, and (bottom) out-of-sample inference. HIPE consistently leads in NLL and small-$q$ inference, while other methods achieve lower RMSE, indicating a trade-off between model calibration and pointwise prediction accuracy.

the center of the search space. However, HIPE maintains its advantage throughout, demonstrating its impact on subsequent BO iterations.

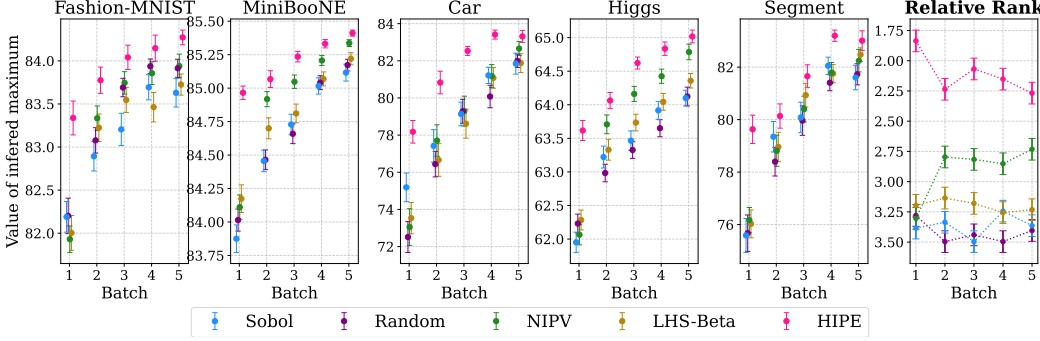

**Figure 7:** Inference regret per method using a batch size $q$ of 8 across LCBench tasks. HIPE is once again the best-performing method overall, but benefits from finding high-valued points during initialization to a greater extent than other methods.

## C.4 Active Learning with Different Batch Sizes

We analyze the effect of batch size in the active learning setting by comparing model performance under small batches ($q = 8$) and large batches ($q = 24$). This evaluation assesses each method's

robustness to changes in batch size and its capability to maintain predictive accuracy and effective hyperparameter learning under varying evaluation budgets.

As shown in Fig. 8, for smaller batches of $q = 8$, HIPE achieves the best performance on the NLL metric and significantly outperforms BALD—the second-best method on NLL—in terms of MSE. In this setting, NIPV and BALD exhibit inconsistent behavior, alternating between the worst performance on MLL and RMSE, respectively.

When the batch size increases to $q = 24$, as visualized in Fig. 8, HIPE continues to perform strongly, consistently ranking among the top two methods across both evaluation metrics. Notably, its relative performance improves with larger batch sizes, particularly in terms of RMSE, indicating that HIPE scales more effectively with increased parallelism in query selection. This suggests that HIPE is better suited for scenarios requiring efficient learning under larger batch evaluations.

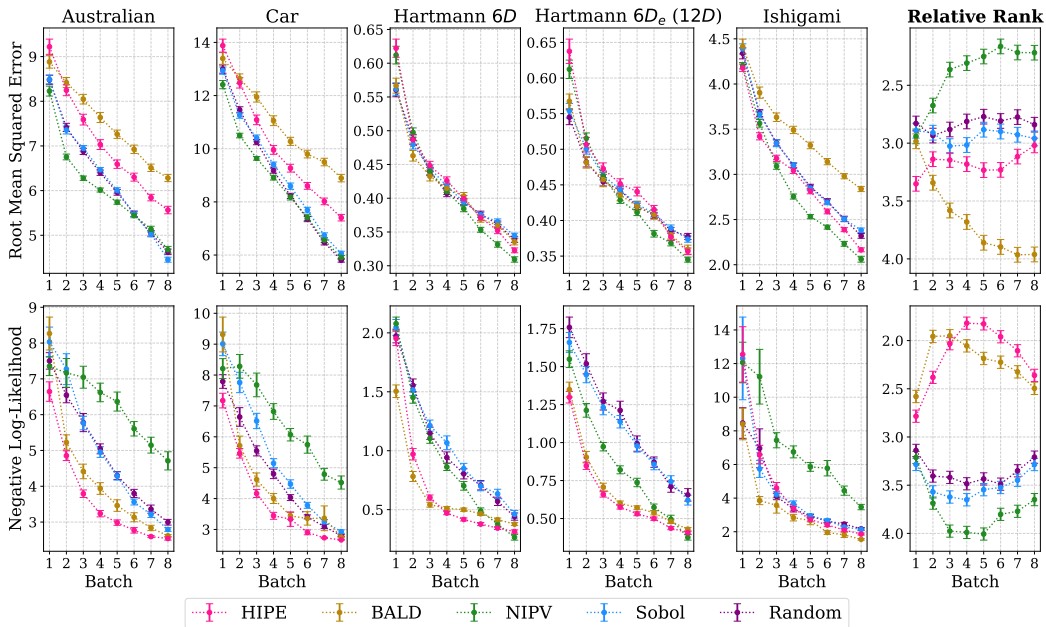

**Figure 8:** Model accuracy results in the batch active learning setting with smaller batches ($q = 8$). RMSE is reported across various synthetic and LCBench surrogate tasks over 8 batches, using 100 random seeds per benchmark. HIPE achieves the best performance on NLL and outperforms BALD, the second-best method on NLL, in terms of RMSE. NIPV and BALD alternate in exhibiting the worst performance on MLL and RMSE, respectively.

## C.5 Complementary Hyperparameter Visualization on SVM

We display the inferred hyperparameter values of the left-out methods (Random, NIPV) on the 40D-SVM. Both methods infer more accurate hyperparameter values than Sobol, but less accurate than HIPE. Compared to Sobol, both Random and NIPV assign smaller lengthscales to the relevant dimensions, indicating improved identification of important features. However, their estimates remain less precise than those obtained by HIPE, which more effectively distinguishes the relevant dimensions, namely the last two.

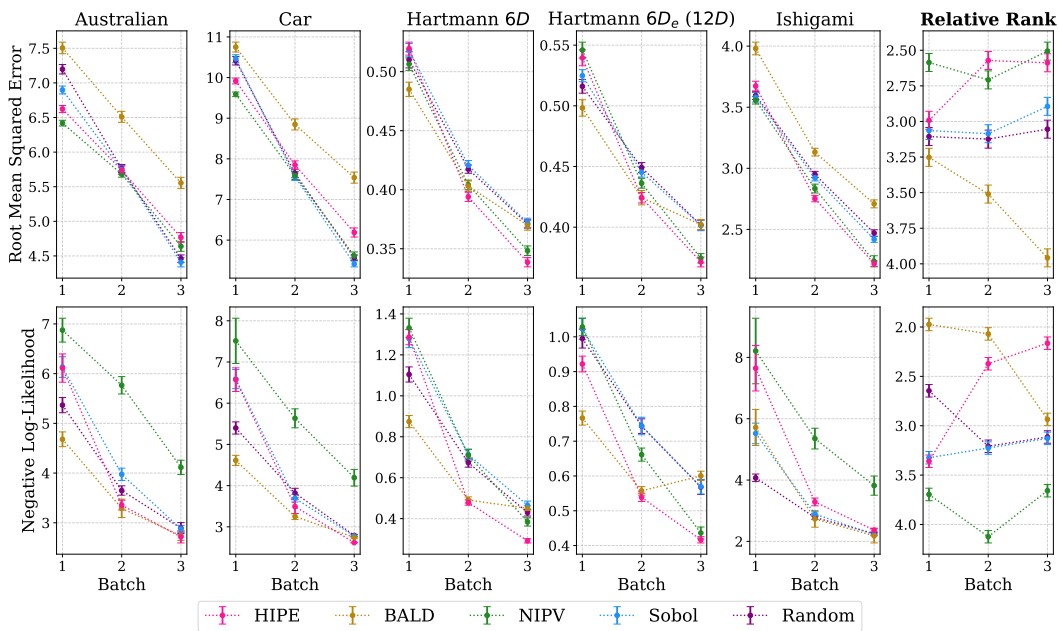

**Figure 9:** Model accuracy results in the batch active learning setting with larger batches ($q = 24$). RMSE is reported across various synthetic and LCBench surrogate tasks over 3 batches, using 100 random seeds per benchmark. HIPE consistently ranks among the top two methods across both metrics, achieving a strong balance between hyperparameter learning and predictive accuracy. With larger batch sizes, HIPE exhibits improved relative performance, particularly on RMSE, demonstrating effective scalability under increased parallel evaluations.

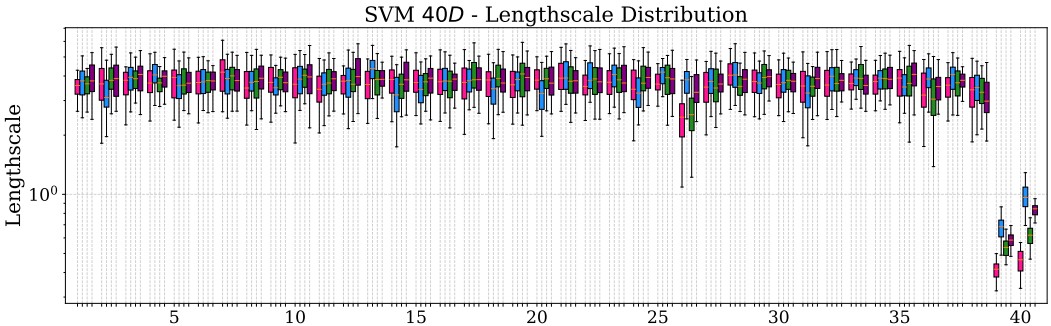

**Figure 10:** Estimated lengthscale hyperparameters for Sobol, Random, NIPV and HIPE on the 40D-SVM task. Random and NIPV infer low lengthscales for the last two dimensions, which are known to be the most important. However, HIPE does so with greater effectiveness.

