# OpenReview forum: "Informed Initialization for Bayesian Optimization and Active Learning"
_NeurIPS.cc/2025/Conference — NeurIPS 2025 poster_

### Official Review · Reviewer_Qt15 · 2025-06-21

**Clarity:** 4
**Significance:** 3
**Originality:** 2
**Rating:** 4
**Confidence:** 3

**Summary:**

The authors propose HIPE, a method for “cold start” initialization of the hyperparameters of surrogate models. The method is intended for application to surrogate models that will be used in the context of sequential learning settings like Bayesian optimization or Bayesian active learning. The proposed acquisition function is an additive combination of two existing acquisition functions, EPIG and BALD.

**Questions:**

- The motivation for HIPE is to balance between hyperparameter learning and space-filling design. Yet the EPIG objective targets reduction in predictive uncertainty, not space-filling design. Under what conditions are these equivalent? If and when they disagree, should the user prefer the design that reduces predictive uncertainty, and why? If not, why not use an acquisition function that optimizes for space-filling design directly?
- I find the notation for the information theoretic measures a bit confusing. In line 79, $y(\boldsymbol{x})$ appears to be defined as a given value of the measurements (a scalar or vector). In eqs. 1, 2 and 4-8, is $y(\boldsymbol{x})$ intended to be a fixed value of the measurements, or the random variable characterizing measurements that arise from a particular value of $\boldsymbol{x}$? The latter is consistent with standard definitions of the EIG. If the latter, should the expectation in eq. 1 be w.r.t. this random variable? If not, could you please clarify how to interpret $\mathrm{H}[y(\boldsymbol{x})]$; what does it mean to take the entropy of a fixed value?
- How are the batches selected? Especially in the large-batch setting, enumerating over every possible batch of locations is generally computationally intractable. It’s common to approximate the optimal batch by greedily optimizing the acquisition function over candidate locations one-at-a-time. How would this practice affect the performance of HIPE?

**Ethical Concerns:**

["NO or VERY MINOR ethics concerns only"]

**Final Justification:**

The paper tackles an important problem, and provides thorough empirical evaluation of the proposed method. My main concerns are (1) limited technical novelty (HIPE is a relatively straightforward combination of two existing methods), and (2) that as currently written, the paper does not make clear for the reader the effect of the EPIG objective in HIPE (to reduce predictive uncertainty, rather than identify space-filling designs). Concern (2) would be resolved by revising the manuscript to include the discussion and clarifications in the authors’ rebuttal. I lean towards acceptance given the value to the Bayesian Optimization community of developing principled approaches for hyperparameter initialization.

**Limitations:**

yes

**Quality:**

3

**Strengths And Weaknesses:**

The paper tackles the important problem of hyperparameter initialization in a principled way. The proposed method is intuitive, and the authors provide thorough empirical evaluation of HIPE against a range of benchmarks and in a variety of settings. Excepting some instances of unclear notation (see Questions below), the paper is clearly written.

The technical contribution strikes me as somewhat limited: HIPE is a relatively straightforward combination of two existing methods (EPIG and BALD). The paper nevertheless carefully considers and specifies the objectives of the initialization phase, which is important and not always straightforward. I find the proposed hyperparameter-free method for setting $\beta$ quite interesting and creative.

---

> ### Author Rebuttal · Authors · 2025-07-30
>
> We thank the reviewer for their review of our work. We appreciate the recognition of the importance of hyperparameter initialization and the clarity of our presentation. Below, we address the specific questions and concerns raised.
>
> &nbsp;
>
> 1. **(...) the EPIG objective targets reduction in predictive uncertainty, not space-filling design. Under what conditions are these equivalent? If and when they disagree, should the user prefer the design that reduces predictive uncertainty, and why? If not, why not use an acquisition function that optimizes for space-filling design directly?**
>
>    Great question. Assuming space-filling refers to criteria like minimax-distance or similar, the EPIG objective and true space-filling designs are most likely never equivalent. Referring to our designs as space-filling is a misnomer on our part, we will adjust this.
>
>    Nonetheless, space-filling or random designs are often employed to globally reduce uncertainty, a view supported by many related works [we will add citations]. If this is indeed the purpose, then such a criterion should be employed directly.
>
>    It is certainly possible—and advisable—to prefer a true space-filling criterion (e.g., minimax) if strict input space diversity is desired. However, this would, by definition of the criteria, be less helpful for *prediction*, which we consider the more important measure for Bayesian Optimization performance.
>
> &nbsp;
>
> 2. **How are the batches selected? Especially in the large-batch setting, enumerating over every possible batch of locations is generally computationally intractable. It’s common to approximate the optimal batch by greedily optimizing the acquisition function over candidate locations one-at-a-time. How would this practice affect the performance of HIPE?**
>
>    All acquisition functions we consider are quasi-Monte Carlo (qMC) acquisition functions, meaning the joint utility (information gain or improvement) is computed as an expectation jointly over samples from \( y(X) \) over a batch of \( q \) candidates. This allows optimization over the full \( q \times D \)-dimensional space, enabling arbitrary batch sizes without enumeration. This general qMC-methodology is described in detail in [1, 2], which we cite in Section 4.2 to explain how HIPE fits into this framework.
>
>    We conducted the full joint optimization using the MC formulation since it was feasible, but a sequential approach to acquisition function optimization would be feasible as well. In low-dimensional ablations, we found that joint optimization yielded slightly larger acquisition function values than sequential, but we do not expect that there would be substantial performance differences between them.
>
> &nbsp;
>
> 3. **I find the notation for the information theoretic measures a bit confusing. In line 79, $y(x)$ appears to be defined as a given value of the measurements (a scalar or vector). In eqs. 1, 2 and 4-8, is $y(x$) intended to be a fixed value of the measurements, or the random variable characterizing measurements that arise from a particular value of $x$? The latter is consistent with standard definitions of the EIG. If the latter, should the expectation in eq. 1 be w.r.t. this random variable? If not, could you please clarify how to interpret $y(x)$; what does it mean to take the entropy of a fixed value?**
>
>    Thank you for carefully reviewing the notation. $y(x)$ and $y(X)$ always denote the random variable characterizing measurements arising from $x$ (or $X$ in the multivariate case). We acknowledge that the notation was overloaded and will clarify this in line 79. The expectation in Eq. 1 is a typo and should indeed be over the random variable $y(x)$. We appreciate the reviewer for spotting this.
>
> &nbsp;
> ____
>
> We hope these clarifications address the reviewer’s outstanding questions and concerns. If not, do not hesitate to ask further clarifying questions. We appreciate the constructive feedback and hope that, if satisfied with our responses, the reviewer might consider raising their score to reflect the strengths and contributions of our work.
>
> ____
> &nbsp;
> **References:**
>
> [1] James T. Wilson, Riccardo Moriconi, Frank Hutter, Marc Peter Deisenroth. *The reparameterization trick for acquisition functions*.
>
> [2] Maximilian Balandat, Brian Karrer, Daniel R. Jiang, Samuel Daulton, Benjamin Letham, Andrew Gordon Wilson, Eytan Bakshy. *BoTorch: A Framework for Efficient Monte-Carlo Bayesian Optimization*.

---

> > ### Comment · Reviewer_Qt15 · 2025-08-01
> >
> > Thank you to the authors for the careful and thorough responses to my questions. As I stated in my original review, I believe the paper tackles an important problem in an intuitive and principled way, which is a significant contribution. I maintain my score, which reflects the balance of this contribution with limited technical novelty.
> >
> > If the paper is accepted, I think it would be made stronger by (i) inclusion of additional discussion of the advantages of each of and differences between space-filling designs and designs which reduce predictive uncertainty, and (ii) revisions to the text to reflect that HIPE targets reduction in predictive uncertainty rather than space-filling design.

---

> > > ### Author Response · Authors · 2025-08-04
> > > **Thanks**
> > >
> > > We thank the reviewer for their feedback, and are happy to see that they find our work worthy of acceptance. We will make sure to better distinguish between space-filling designs and designs that reduce predictive uncertainty in the paper. Moreover, we will clarify that HIPE targets the latter.
> > >
> > > Thanks once again,
> > >
> > > The authors

---

### Official Review · Reviewer_gQSM · 2025-06-29

**Clarity:** 4
**Significance:** 3
**Originality:** 3
**Rating:** 5
**Confidence:** 3

**Summary:**

The paper introduces a new acquisition function for Bayesian Optimization and Active Learning in the case of hyperparameter uncertainty modeled via fully Bayesian Gaussian Processes. This acquisition function is designed to perform especially well in the large-batch and few shot case. It balances hyperparameter uncertainty reduction, as well as space filling designs for uniform test space distributions.

**Questions:**

- Why don't you take Query-By-Committee and its variant (Riis et al., 2022) as a baseline since it outperforms BALD in the settings used in their paper ?
- Would Statistical distance-based Active Learning from Hvarfner et al., 2023 be another choice as a comparative acquisition function?
- I am wondering whether there are any theoretical guarantees to proof for your acquisition function

**Ethical Concerns:**

["NO or VERY MINOR ethics concerns only"]

**Final Justification:**

Increased score from Borderline accept to accept due to the clarifications and additional experiments given by the authors (see rebuttal for more information).

**Limitations:**

Yes

**Quality:**

3

**Strengths And Weaknesses:**

- The topic of hyperparameter uncertainty in practical examples is under-adressed in the current GP literature and therefore of great interest
- The paper is well written and gives a good intuition for the new acquisition function
- The experiments are extensive and support the superiority of the new acquisition function against baselines

- I am not sure, whether the authors compare to the current state of the art in their experiments (see questions for suggestions)

---

> ### Author Rebuttal · Authors · 2025-07-30
>
> We thank the reviewer for their insightful and constructive feedback. We appreciate the recognition of the importance the topic, and the quality of the presentation. Below, we address the specific questions and suggestions raised.
>
> 1. **Why don't you take Query-By-Committee and its variant (Riis et al., 2022) as a baseline since it outperforms BALD in the settings used in their paper?**
>
>    While BQBC outperforms BALD in the Riis et al. paper, it does not do so in the Hvarfner et al. paper, where BALD performs substantially better on identical benchmarks. BQBC performs identically across the two papers. We suspect there may have been a bug in the Riis et al. implementation of BALD, since it has subsequently been more competitive. Therefore, we consider our choice of BALD baseline appropriate.
>
> &nbsp;
>
> 2. **Would Statistical distance-based Active Learning from Hvarfner et al., 2023 be another choice as a comparative acquisition function?**
>
>    Statistical distance-based Active Learning (SAL) is indeed a strong candidate in the sequential setting but as far as we know, _it has not been tested in a batch setting_. However, we have run the version that is publicly available in BoTorch. This variant is runnable as a batched acquisition function. Below, you can see updated AL task performance with SAL included:
>
>    MLL performance (larger is better): HIPE performs best, followed by BALD and SAL, which are both competitive.
>
>    | Function   | BALD            | HIPE            | NIPV            | Random          | SAL             | Sobol           |
>    |------------|-----------------|-----------------|-----------------|-----------------|-----------------|-----------------|
>    | Australian | -2.66 ± 0.08    | **-2.61 ± 0.22**| -7.47 ± 0.80    | -3.00 ± 0.17    | -4.11 ± 0.73    | -3.40 ± 0.24    |
>    | car        | -3.07 ± 0.45    | **-2.68 ± 0.02**| -6.09 ± 0.65    | -3.09 ± 0.14    | -4.16 ± 0.82    | -3.30 ± 0.27    |
>    | hartmann6  | -0.48 ± 0.02    | **-0.31 ± 0.02**| -0.43 ± 0.06    | -0.57 ± 0.07    | -0.48 ± 0.02    | -0.63 ± 0.07    |
>    | ishigami   | **-1.74 ± 0.09**| -2.59 ± 0.23    | -6.94 ± 1.26    | -2.45 ± 0.17    | -2.50 ± 0.42    | -2.32 ± 0.12    |
>
> &nbsp;
>
>    RMSE performance: NIPV generally performs best. Batched SAL does not perform well on RMSE.
>    | Function   | BALD            | HIPE            | NIPV            | Random          | SAL             | Sobol           |
>    |------------|-----------------|-----------------|-----------------|-----------------|-----------------|-----------------|
>    | Australian | 6.40 ± 0.15     | 5.29 ± 0.11     | 4.96 ± 0.12     | **4.70 ± 0.10** | 6.42 ± 0.14     | 4.91 ± 0.11     |
>    | car        | 8.67 ± 0.20     | 6.99 ± 0.17     | **5.99 ± 0.15** | 6.06 ± 0.13     | 8.96 ± 0.22     | 6.23 ± 0.15     |
>    | hartmann6  | 0.37 ± 0.01     | 0.34 ± 0.01     | **0.34 ± 0.01** | 0.37 ± 0.01     | 0.37 ± 0.01     | 0.37 ± 0.01     |
>    | ishigami   | 2.78 ± 0.05     | **2.25 ± 0.04** | 2.25 ± 0.06     | 2.53 ± 0.05     | 2.93 ± 0.05     | 2.47 ± 0.04     |
>
> Thanks for the suggestion to include SAL, we will happily include it in the results for the CR.
>
> &nbsp;
>
> 3. **I am wondering whether there are any theoretical guarantees to proof for your acquisition function.**
>
>    Our acquisition function is designed to produce a well-informed initialization to improve subsequent Bayesian Optimization performance rather than to minimize regret immediately. As such, regret bounds are not currently available. While theoretical guarantees on model predictive performance might be possible, such results are rare in the broader Bayesian active learning literature. We acknowledge this as a limitation and an interesting direction for future work.
>
> &nbsp;
> ___
> We hope these clarifications address the reviewer’s questions and concerns. We appreciate the constructive feedback and hope that, if satisfied with our responses, the reviewer might consider raising their score to reflect the strengths and contributions of our work.

---

> > ### Comment · Reviewer_gQSM · 2025-08-04
> >
> > Dear authors,
> >
> > thank you for the clarification and extensive rebuttal. Therefore, I am willing to raise my score and recommend the acceptance of the paper.

---

### Official Review · Reviewer_XxAJ · 2025-06-30

**Clarity:** 2
**Significance:** 3
**Originality:** 2
**Rating:** 4
**Confidence:** 3

**Summary:**

The authors introduce a novel acquisition function (HIPE) for Bayesian Optimization (BO) that aims for an effective initialization of BO algorithms. HIPE incorporates surrogate model hyperparameters into the selection of new candidates to account for different importances of dimensions while aiming to be space-filling (i.e., achieving a good coverage of candidate configurations). They empirically evaluate the proposed acquisition function on BO tasks, showing HIPE's effectiveness.

**Questions:**

- in Fig. 2, the rank of BALD and NIPV tremendously changes when considering RMSE instead of NLL. Can the authors explain why this is happening?

**Ethical Concerns:**

["NO or VERY MINOR ethics concerns only"]

**Final Justification:**

The authors have addressed my main concerns and have added additional results as requested. Overall the authors provide an incremental, but still valuable contribution.

**Limitations:**

yes

**Quality:**

3

**Strengths And Weaknesses:**

**Strengths**

- The work tackles the important problem of making batch BO more sample-efficient by better initialization/selection of query points
- mostly clearly written
- The proposed approach is simple and easy to adopt
- extensive empirical evaluation is provided

**Weaknesses**

- as far as I understood, the novel acquisition function proposed is a linear combination of EPIG and BALD. Therefore, it is not necessary to discuss NIPV in Sec. 2
- Fig. 1: it is not clear what the authors consider to be "optimal" in this example. It is not easy to see why HIPE selects better points than NIPV since both have broad coverage.
- The authors do not explain why strategies like Sobol are not able to accomplish a space-filling selection (see Fig. 1 caption).
- Prop. 1 introduces a connection between EPIG, BALD, and HIPE. However, it is not clear to me what this connection means for the proposed method. For example, does this influence the convergence properties of the BO? It would be good if the authors could comment on this.
- A theoretical consideration of HIPE in terms of convergence would be good in general.
- in line 209, the authors claim that "entropy in Eq. 10 is Gaussian". Can the authors provide proof or a reference showing that this is the case?
- in the experimental section, it would be great if the authors could provide an ablation varying the number of batches and the batch size to see if HIPE performs well for different settings as well (esp. for Sec. 5.2, 5.3, 5.4).
- to further strengthen the evaluation, considering other HPO-benchmarks like [1] and [2] could be beneficial.

**References**

[1] HPO-B: A Large-Scale Reproducible Benchmark for Black-Box HPO based on OpenML. Arango et al. 2021.

[2] HPOBench: A Collection of Reproducible Multi-Fidelity Benchmark Problems for HPO. Eggensperger et. al. 2021.

---

> ### Author Rebuttal · Authors · 2025-07-30
>
> We thank the reviewer for their detailed and thoughtful review. We appreciate the recognition of the importance of the method, as well as its simplicityekecnerbflrktggenuicujfrfttluvnj, the clarity of our presentation, and the extensive empirical evaluation. Below, we address the specific concerns raised.
>
> &nbsp;
>
> 1. **As far as I understood, the novel acquisition function proposed is a linear combination of EPIG and BALD. Therefore, it is not necessary to discuss NIPV in Sec. 2.**
>
>    It is correct that our method is a combination of EPIG and BALD. We introduced NIPV because it is strongly related to EPIG and is arguably the more well-known method in the broader active learning literature.
>
> &nbsp;
>
> 2. **It is not clear what the authors consider to be ‘optimal’ in this example (Fig. 1). It is not easy to see why HIPE selects better points than NIPV since both have broad coverage.**
>
>    We try to motivate this in the text accompanying the figure but agree it should be clearer in the figure itself. The coverage is basically equally good for NIPV and HIPE, but HIPE additionally has the hyperparameter-informative axis alignment like BALD. Thus, by design, HIPE intuitively should be better for hyperparameter learning than NIPV while visually retaining equal coverage.
>
>    We will add helper lines to the figure to emphasize the axis alignment in HIPE.
>
>    Figure 1 is meant primarily as an illustration to provide intuition of the typical behavior of the different initialization approaches; in general it is not straightforward to draw conclusions on the performance without actually evaluating performance (in terms of model fit and/or optimization performance).
>
> &nbsp;
>
> 3. **Prop. 1 introduces a connection between EPIG, BALD, and HIPE. However, it is not clear what this connection means for the proposed method. For example, does this influence the convergence properties of the BO?**
>
>    Proposition 1 shows that, for a certain choice of beta, HIPE is equivalent to jointly maximizing both EPIG and BALD, justifying the linear combination from a theoretical perspective. However, this has no direct influence on subsequent BO convergence. A formal proof of convergence for BO with dynamically updated hyperparameters (e.g., MAP or fully Bayesian estimates) is still an open problem and beyond the scope of this work.
>
> &nbsp;
>
> 4. **In line 209, the authors claim that ‘entropy in Eq. 10 is Gaussian’. Can the authors provide proof or a reference showing that this is the case?**
>
>    Our original formulation was imprecise. More precisely, $y(x)∣X,\theta,{D}$ is the predictive posterior at input $x$ for a Gaussian Process with data $D$. This predictive posterior is Gaussian, and the entropy of a Gaussian has a closed-form expression. Since the entropy depends only on the covariance, it is easily computable. This is a standard result in Gaussian Process literature (e.g., Rasmussen and Williams, 2005).
>
> &nbsp;
>
> 5. **It would be great if the authors could provide an ablation varying the number of batches and the batch size to see if HIPE performs well for different settings as well (esp. for Sec. 5.2, 5.3, 5.4).**
>
>    We have included the terminal performance for 5 batches with a batch size of 8 (initialization in the first batch only) as a table below; We will also include them in a potential CR. We can add additional ones as plots in the Appendix, similar to the active learning ablations. HIPE performs very well in this slightly smaller batch size, but one can naturally expect the importance of initialization to be largest when it constitutes a larger part of the overall budget.
>
>    | Function       | HIPE            | LHS-Beta        | NIPV            | Random          | Sobol           |
>    | -------------- | --------------- | --------------- | --------------- | --------------- | --------------- |
>    | Fashion-MNIST  | **84.65 ± 0.07**| 84.19 ± 0.06    | 84.29 ± 0.08    | 84.17 ± 0.04    | 84.23 ± 0.04    |
>    | MiniBooNE      | **85.47 ± 0.02**| 85.25 ± 0.05    | 85.40 ± 0.02    | 85.24 ± 0.04    | 85.21 ± 0.06    |
>    | car            | **84.38 ± 0.15**| 83.10 ± 0.36    | 83.74 ± 0.19    | 82.36 ± 0.37    | 82.98 ± 0.33    |
>    | higgs          | **65.21 ± 0.08**| 64.56 ± 0.13    | 64.90 ± 0.11    | 64.18 ± 0.14    | 64.29 ± 0.14    |
>    | segment        | **83.86 ± 0.18**| 83.12 ± 0.14    | 83.39 ± 0.12    | 82.70 ± 0.18    | 82.95 ± 0.24    |
>
>    Hopefully, this provides evidence that HIPE does well even with smaller batch sizes.
>
> &nbsp;
>
> 6. **To further strengthen the evaluation, considering other HPO-benchmarks like [1] and [2] could be beneficial.**
>
>    We thank the reviewer for these references and will consider including these benchmarks in future evaluations to broaden the empirical validation.
>
> &nbsp;
>
> 7. **in Fig. 2, the rank of BALD and NIPV tremendously changes when considering RMSE instead of NLL. Can the authors explain why this is happening?**
>
>    Certainly. Since NIPV spreads out points to maximally reduce integrated variance, hyperparameter estimation gets difficult. On the contrary, BALD tends to stack axis-aligned queries (Fig. 1) making hyperparameter estimation simple at the cost of poor coverage.
>
>    With great coverage, the mean prediction (as with NIPV) is generally fairly accurate in most places, but poor hyperparameter estimation simultaneously can lead to drastic over- and underconfidence that punishes the MLL. BALD is the opposite: rarely over-or underconfident, but the mean predictions are not very accurate.
>
> &nbsp;
> ____
> We hope these responses address the reviewer’s concerns. We appreciate the constructive feedback and hope that, if satisfied with our responses, the reviewer might consider raising their score to better reflect the strengths and contributions of our work.

---

> > ### Comment · Reviewer_XxAJ · 2025-08-05
> >
> > I thank the authors for the clarifications and additional results.
> >
> > My points have been addressed appropriately; therefore, I will increase my score to 4.
> >
> > However, I strongly recommend adding a convergence analysis for the proposed acquisition function. This shouldn't be too difficult as HIPE is a linear combination of BALD and EPIG (at least showing that HIPE cannot be worse than BALD and EPIG should be straightforward).Such a result would significantly increase the work's value.

---

### Official Review · Reviewer_g6zV · 2025-07-01

**Clarity:** 3
**Significance:** 3
**Originality:** 3
**Rating:** 5
**Confidence:** 3

**Summary:**

In this work, the authors propose a new initialisation scheme for sequential experimental design methods, specifically for active learning and Bayesian Optimisation. The authors argue that previously proposed initialisation schemes may not be sufficiently space-filling and that they often do not sufficiently consider how to learn hyperparameters of the surrogate model. Based on this the authors propose hyperparameter-informed predictive exploration (HIPE) which essential combines initialisation/sampling strategies that either optimise for predictive information gain or hyperparameter learning. The initial formulation of HIPE contains a hyperparameter that should balance either optimising predictive information gain or hyperparameter learning which can be cumbersome to tune at run time. However, the authors show that this hyperparameter can be found optimally and precomputed without traditional hyperparameter tuning. The authors then illustrate the efficacy of the proposed method via a number of experiments.

**Questions:**

I firstly list larger questions here, and below more optional questions.

My main questions in relation to aforementioned weaknesses and clarifications, which if most of were addressed/discussed would cause me to raise my score:
1. The authors discuss how the presented method may be too expensive for some applications - would the authors consider including compute times in the work to make it clear how expensive the method is?
2. Am I understanding correct that using the proposed method requires a candidate set of initial data points?
3. I think some ablation studies and some more insights on experimental results would be useful. I elaborate here:
    - Do the authors have any insights on why the init scheme seems to be relatively problem sensitive? In Figure 2, 3 and 4 there are some problem types where HIPE perform a lot worse than in other problems. Some discussion on why this might be the case would be very useful for practitioners and further development.
    - Is there any reasons the authors only show the Sobol results in Figure 5 (right) and not all the other methods?
    - Did the authors look at performance of their proposed method in the BO setting for iterations > 2?
4. Do the authors have any insights on how the proposed init scheme would work for other surrogate models such as BNNs or Random Forests?
5. I think the title is a little confusing (although this is an opinion) whilst I think the abstract is overclaiming. To me the title indicates something where the init scheme is prior knowledge dependant, but HIPE is actually "fully automized" requiring no prior from the user - perhaps including Hypeparameter in the title as in the method would be helpful. In the abstract, the authors write that HIPE outperforms other standard init schemes, which is not always the case - I believe this should be amended.

Minor comments or questions:
1. There is a citation missing in relation to BO surrogate quality and calibration in lines 34 and lines 136-139, see [1].
2. The authors write "at the core of BO is an acquisition function", but I would argue that the surrogate is equally important.
3. In line 162 the authors write that Scrambled Sobol sequences fail to achieve the desired coverage, but I do not see any citation providing evidence for this.
4. In lines 163-166 the authors repeat much of what was already said in the paragraph running from lines 149-158 if I am not mistaken.
5. Based on Figure 1, I have a hard time seeing qualitative differences between NIPV and HIPE. Maybe the authors could discuss this more in detail.
6. In line 306 there appears to be a grammatical error in the sentence beginning with "its HIPE".

[1]: Jonathan Foldager, Mikkel Jordahn, Lars Kai Hansen, and Michael Riis Andersen. On the role of
model uncertainties in Bayesian optimization. In Uncertainty in Artificial Intelligence (UAI), 2023.

**Ethical Concerns:**

["NO or VERY MINOR ethics concerns only"]

**Final Justification:**

Increased score from borderline accept to accept based on informative replies in rebuttal. See comment to authors for details.

**Limitations:**

Yes

**Quality:**

3

**Strengths And Weaknesses:**

Strengths
- The paper reads clearly and is motivated well.
- There are essentially _no_ hyperparameters in the method which is a big plus for the sequential experimental design setting.
- Good initialisation schemes for sequential experimental design setting is an often overlooked problem that in practice can translate to big differences in performance - this paper contributes significantly to this issue.

Weaknesses
- Missing run time costs of experiments which can be important for these types of applications.
- The method is presented only for GPs, which may be a shortcoming as BNNs and NNs are becoming more used surrogates in the sequential experimental design setting.
- The experimental setup and evaluation is somewhat strict and specific in some senses. I elaborate below.

---

> ### Author Rebuttal · Authors · 2025-07-30
>
> We thank the reviewer for their detailed feedback and positive evaluation of our work. We appreciate that the reviewer finds our paper to read well, and that they recognize the potential significance and historical lack of attention for our problem setting. Below, we address each of the weaknesses and questions raised.
>
> *&nbsp;*
>
> 1. **The authors discuss how the presented method may be too expensive for some applications - would the authors consider including compute times…**
>
> Certainly. Below is a runtime table for $q=8$ and $N=32$, namely 4 batches of 8 evaluations each with the first batch being initialization. We show runtime across 20 seeds for HIPE, NIPV and random on Hartmann (6D). The "BO loop" components are summed across all batches, meaning that "Model fit during BO loop" involves 3 unique model fittings for the three BO batches. All experiments are run on one A40 GPU. In our experience, this makes acquisition function optimization substantially faster, but accelerates model fitting less. HIPE is slower than NIPV and Random, but the runtime of initialization is still less than 20% of the overall runtime of BO in this setup. Naturally, the proportion of the total runtime allocated to initialization increases with fewer total batches.
>
> | Runtime Component (sec)      | HIPE          | NIPV          | Random        |
> |------------------------------|---------------|---------------|---------------|
> | Initial model fit / sampling | 21.91 ± 0.15  | 21.73 ± 0.18  | 0.02 ± 0.00   |
> | AcqOpt during init           | 19.61 ± 1.12  | 8.62 ± 0.37   | —             |
> | Model fit during BO loop     | 181.79 ± 7.60 | 189.90 ± 5.17 | 177.33 ± 7.69 |
> | AcqOpt during BO loop        | 4.97 ± 0.18   | 4.44 ± 0.05   | 4.35 ± 0.06   |
> | Objective evals (synthetic)  | 0.00 ± 0.00   | 0.00 ± 0.00   | 0.00 ± 0.00   |
>
> Since we use a fully Bayesian surrogate with NUTS and large batch sizes, all steps are relatively time-consuming. Fully Bayesian models tend to be expensive in general, and HIPE shares similar computational demands. We want to emphasize though that in the few-shot settings considered in this work this cost is typically dwarfed by the cost of running evaluations, which in practice can often be on the order of days or weeks (e.g. training large ML models or running online A/B tests).
>
>
> ***&nbsp;***
>
> 2. **Am I understanding correct that using the proposed method requires a candidate set of initial data points?**
>
> Our method does not require a candidate set of initial data points. What it does require / allow for is a distribution $p(x_*)$ to inform the method about promising regions (e.g. based on user priors or domain knowledge). In the absence of prior knowledge about promising regions, a uniform distribution provides the best coverage, which is what we used in all experiments. However, users *can* bias initialization toward a-priori promising regions with a non-uniform $p(x_*)$. From an implementation perspective, we represent $p(x_*)$ as an empirical distribution of sample points. Importantly though, these sample points *do not* need to be evaluated on the objective function.
>
> ***&nbsp;***
>
> 3. **Is there any reasons the authors only show the Sobol results in Figure 5 (right) and not all the other methods?**
>
> Only to de-clutter the plot. Since there were 40 dimensions to visualize, we thought we would only include the two most relevant comparisons. The other methods are very similar to Sobol.
>
> **&nbsp;**
>
> 4. **Did the authors look at performance of their proposed method in the BO setting for iterations > 2?**
>
> Yes, we have included the terminal performance for 5 batches with a batch size of 8 (initialization in the first batch only) as a table below; We will also include them in a potential CR. HIPE performs very well in this slightly smaller batch size, but one can naturally expect the importance of initialization to be largest when it constitutes a larger part of the overall budget.
>
> | Function       | HIPE            | LHS-Beta        | NIPV            | Random          | Sobol           |
> | -------------- | --------------- | --------------- | --------------- | --------------- | --------------- |
> | Fashion-MNIST  | **84.65 ± 0.07**| 84.19 ± 0.06    | 84.29 ± 0.08    | 84.17 ± 0.04    | 84.23 ± 0.04    |
> | MiniBooNE      | **85.47 ± 0.02**| 85.25 ± 0.05    | 85.40 ± 0.02    | 85.24 ± 0.04    | 85.21 ± 0.06    |
> | car            | **84.38 ± 0.15**| 83.10 ± 0.36    | 83.74 ± 0.19    | 82.36 ± 0.37    | 82.98 ± 0.33    |
> | higgs          | **65.21 ± 0.08**| 64.56 ± 0.13    | 64.90 ± 0.11    | 64.18 ± 0.14    | 64.29 ± 0.14    |
> | segment        | **83.86 ± 0.18**| 83.12 ± 0.14    | 83.39 ± 0.12    | 82.70 ± 0.18    | 82.95 ± 0.24    |
>
> ***&nbsp;***
>
> 5. **The method is presented only for GPs, which may be a shortcoming as BNNs and NNs are becoming more used surrogates.**
>
> We agree that Bayesian Neural Networks (and other types of surrogates) are increasingly used, though less so for Random Forests. HIPE can be applied to BNNs, where uncertainty over *θ* corresponds to uncertainty over all neural network parameters. However, the computation is more involved than for GPs, as BNNs lack closed-form expressions for posterior variance or conditioning on new data.
>
> Random Forests do not inherently have a Bayesian treatment of model parameters and thus cannot use HIPE directly.
>
> ***&nbsp;***
>
> 6. **Do the authors have any insights on why the init scheme seems to be relatively problem sensitive?**
>
> Yes, we do. Since HIPE’s EPIG criterion minimizes integrated (average) variance over the search space, it tends to avoid points near boundaries, as average variance is rarely reduced by sampling near edges. Sobol and random methods have no such bias, which explains the performance swings between HIPE and random methods. Differences between NIPV and HIPE are less straightforward.
>
> For some problems, HIPE’s more centered queries are beneficial early on, while for others (e.g., SVM), where optima lie near boundaries, this is less so. However, HIPE compensates by correctly identifying active dimensions and catches up in later batches.
>
> ***&nbsp;***
>
> 7. **I think the title is a little confusing...**
>
> Does the reviewer refer to the term “*Informed”?* We are open to suggestions on what a more representative title would be.
>
> ***&nbsp;***
>
> 8. **...whilst I think the abstract is overclaiming.**
>
> HIPE is designed to produce the most informed model (in terms of point spread and hyperparameter inference) to enable successful subsequent BO batches, rather than to achieve the best performance after batch 1. From this perspective, we believe the statement is accurate but agree that we can make it more precise to improve clarity.
>
> ***&nbsp;***
> ___
>
> Minor comments have been addressed and will be included in the camera-ready version:
>
> 1. The missing citation related to BO surrogate quality and calibration has been added.
> 2. We agree that the surrogate model is equally important as the acquisition function in BO; this is now reflected in line 80.
> 3. The claim about Scrambled Sobol sequences not achieving desired coverage anecdotal. We will avoid general statements and discuss Fig. 1 specifically.
> 4. We have rewritten the repeated content in lines 163-166.
> 5. The discussion around Figure 1 has been expanded to better highlight qualitative differences between NIPV and HIPE, including a modified figure clarifying axis alignment.
> 6. The grammatical error in line 306 has been corrected.
>
> &nbsp;
> ___
>
> We once again thank the reviewer for their constructive feedback and appreciate their recognition of the paper’s clarity, motivation, and the relevance of the problem setting. We hope that our responses have addressed the reviewer’s questions regarding runtime analysis and ablation studies.
>
> In light of these additions, we hope the reviewer finds the additional results compelling and the overall submission strengthened.

---

> > ### Comment · Reviewer_g6zV · 2025-08-04
> >
> > Dear authors
> >
> > Thank you very much for your thorough rebuttal. I sincerely appreciate the answers, and believe the paper should be accepted, which is why I have raised my score. My final comments would be to include the following in the paper:
> >
> > 1. Whilst I agree that runtimes often is not an issue, there are applications where BO loops need to be fast (e.g. live instrument calibration, model predictive control), and I therefore think it is relevant to include the runtimes in the final paper.
> > 2. Perhaps it is just me, but it was not clear to that one could further bias $p(x_*)$, in light of which the title of the paper makes more sense. I do however think that perhaps the title still sells the method short, as it is clear from the experiments that one does not _have_ to make the inits informed for the init scheme to work well - but this is just an opinion.
> > 3. I would suggest including the additional plots in the appendix. As a reader, I found it a bit strange that only one other method was compared to (and it honestly appeared a bit like cherry-picking).
> > 4. I think the comments on why HIPE performs worse on some problems would be very useful to include in the discussion section, and hope this can be done for a CR version.
> >
> > Once again, thank you for your reply, and congratulations on a nice paper.

---

> > > ### Author Response · Authors · 2025-08-04
> > > **Thanks**
> > >
> > > Thanks to the reviewer for their further comments. We greatly appreciate that you find the paper worthy of acceptance.
> > >
> > > 1. Certainly. We agree, and will make sure to include it in the CR.
> > > 2. That is fair, we will keep it in mind!
> > > 3. We will include HP visualizations for other methods in the appendix as well.
> > > 4. Absolutely, we will include it in the discussion.
> > >
> > > Thanks once again.
> > >
> > > Best,
> > > The authors

---

### Official Review · Reviewer_MUHR · 2025-07-03

**Clarity:** 2
**Significance:** 2
**Originality:** 2
**Rating:** 3
**Confidence:** 3

**Summary:**

This paper introduces Hyperparameter-Informed Predictive Exploration (HIPE), a principled initialization strategy for Bayesian Optimization and active learning that jointly balances space-filling exploration with efficient hyperparameter learning using information-theoretic objectives. The method is proposed for the large-batch and few-shot setting.

**Questions:**

1. Could the authors justify why setting $\beta = $EIG$(y(x_*); \theta)$ achieves a balance between the two competing objectives?

2. What do the red and blue boxes represent in the right subplot of Figure 5? A legend or explanation would improve clarity.

**Ethical Concerns:**

["NO or VERY MINOR ethics concerns only"]

**Final Justification:**

The authors have addressed most of my concerns. However, I still think that the proposed method lacks novelty, as the new acquisition is essentially a linear combination of BALD and EPIG. While a combination approach is not necessarily a problem, the paper does not convincingly demonstrate that such a simple combination yields a significant improvement in most of the experiments. Moreover, even though the authors explain why $\beta=EIG$ is suggested, I still feel this choice to be rather heuristic and not theoretically grounded.

**Limitations:**

Yes.

**Paper Formatting Concerns:**

No.

**Quality:**

2

**Strengths And Weaknesses:**

**Strengths**:

1. The paper explores an interesting direction direction about the initialization for Bayesian Optimization and Active Learning in few-shot, large-batch scenarios.

2. The experimental evaluation spans multiple settings, deterministic and noisy test functions and hyperparameter tuning, including LCBench and SVM.

**Weaknesses**:

1. The core idea behind HIPE offers limited novelty, essentially combining two well-established acquisition strategies, and introduces a hyperparameter $\beta$ that may be difficult to tune. Moreover, the fast implementation of the objective builds heavily on existing techniques.

2. While the authors introduce NIPV, it is not empirically evaluated in combination with BALD, leaving its practical benefits underexplored. The authors are encouraged to also test this combination.

3. Aside from Proposition 1, which establishes an information-gain equivalence, the paper lacks theoretical guarantees (e.g., regret analysis) to support the proposed method’s performance.

4. The current experiment presentation lacks sufficient details to enable reproducibility, particularly regarding the choice and tuning of the hyperparameter $\beta$, which is not explicitly specified.

---

> ### Author Rebuttal · Authors · 2025-07-30
>
> We would like to thank the reviewer for their feedback on our work. Moreover, we are appreciative that the reviewer acknowledges the relevance of our problem setting. Regarding the design of the hyperparameter $\beta$, we would like to bring to the reviewer’s attention that we provided a method for setting it in the paper, in Section 4.1, L192. All our responses are outlined below.
>
>
> 1. **The current experiment presentation lacks sufficient details to enable reproducibility, particularly regarding the choice and tuning of the hyperparameter beta, which is not explicitly specified.**
>
>    We use a principled approach to set the hyperparameter beta, as described at line 192 in the paper, and apply this consistently throughout all experiments. Appendix A1 provides detailed information on the experimental setup, including hyperpriors and software, to support reproducibility. Our full experimental code is available on GitHub, the link to which is in Appendix A1 as well. We are happy to provide further clarifications if needed.
>
> &nbsp;
>
> 2. **Could the authors justify why setting β = EIG achieves a balance between the two competing objectives?**
>
>
>    The EPIG and BALD objectives are generally on very different scales, making weighting necessary. Setting $β = \text{EIG(}y(x_*); \theta)$ provides a principled weighting that reflects how much hyperparameter information actually translates to predictive information on the test set. This choice ensures that the hyperparameter learning component (BALD) is weighted according to its actual relevance for prediction, rather than using an arbitrary constant. When hyperparameter uncertainty has little impact on test predictions, $\beta$ will be small, emphasizing space-filling. When hyperparameter uncertainty significantly affects predictions, $\beta$ will be larger, emphasizing hyperparameter learning. This adaptive weighting achieves the desired balance automatically without manual tuning.
>
> &nbsp;
>
> 3. **While the authors introduce NIPV, it is not empirically evaluated in combination with BALD, leaving its practical benefits underexplored.**
>
>    We acknowledge that NIPV + BALD can have practical value on its own. However, NIPV is not combined with BALD because it does not adhere to an information-theoretic framework like HIPE. Consequently, the same derivation as in Prop. 1 cannot be made, nor can beta be derived in a similar manner to achieve the desired trade-off between hyperparameter learning and space-filling. Given this, combining the two would be more heuristic than principled.
>
> With that said, We have included the NIPV + BALD combination in the table below. While it wins on 1/5 LCBench problems, the BALD term generally dominates the NIPV term, leading to under-exploration.
>
>    | Function       | HIPE            | NIPV + BALD     | LHS-Beta        | NIPV            | Random          | Sobol           |
>    | -------------- | --------------- | --------------- | --------------- | --------------- | --------------- | --------------- |
>    | Fashion-MNIST  | 84.06 ± 0.09    | **84.15 ± 0.07**| 83.77 ± 0.08    | 83.45 ± 0.09    | 83.75 ± 0.09    | 83.62 ± 0.08    |
>    | MiniBooNE      | **85.37 ± 0.02**| 85.31 ± 0.03    | 85.25 ± 0.03    | 85.27 ± 0.03    | 85.20 ± 0.04    | 85.21 ± 0.03    |
>    | car            | **83.34 ± 0.24**| 82.99 ± 0.21    | 82.91 ± 0.26    | 82.31 ± 0.33    | 82.36 ± 0.27    | 80.95 ± 0.56    |
>    | higgs          | **65.17 ± 0.04**| 64.91 ± 0.04    | 64.77 ± 0.06    | 64.99 ± 0.05    | 64.61 ± 0.08    | 64.54 ± 0.08    |
>    | segment        | **81.55 ± 0.28**| 81.09 ± 0.29    | 81.31 ± 0.24    | 80.81 ± 0.31    | 81.01 ± 0.30    | 80.92 ± 0.29    |
>
> &nbsp;
>
> 4. **Aside from Proposition 1, which establishes an information-gain equivalence, the paper lacks theoretical guarantees (e.g., regret analysis) to support the proposed method's performance.**
>
>    Our acquisition function is designed to produce a well-informed initialization to improve subsequent Bayesian Optimization performance rather than to minimize regret immediately. As such, regret bounds are not currently available. While theoretical guarantees on model predictive performance might be possible, such results are rare in the broader Bayesian active learning literature. We acknowledge this as a limitation and an interesting direction for future work.
>
> &nbsp;
>
> 5. **What do the red and blue boxes represent in the right subplot of Figure 5?**
>
>    The red and blue boxes in Figure 5 (right) represent the interquartile ranges of the estimated lengthscale distributions for different initialization methods. The red boxes show lengthscales estimated after HIPE initialization, while the blue boxes show lengthscales after Sobol initialization. The plot demonstrates that HIPE correctly identifies the last two dimensions (the SVM's global regularization parameters) as most important by assigning them substantially smaller lengthscales compared to Sobol. We will add a legend to clarify this in the camera-ready version.
>
> &nbsp;
>
> ---
> We once again thank the reviewer for their service, and hope our rebuttal addresses the reviewer’s questions and concern regarding how the hyperparameter $\beta$ is automatically set. Should the reviewer have any further questions or require additional details, we would be more than happy to provide them.

---

> ### Comment · Reviewer_MUHR · 2025-08-05
>
> Thanks the authors to provide the additional experiments and detailed explanation to my concerns. I would raise my score to 3.
>
> However, I still think that the proposed method lacks novelty, as the new acquisition is essentially a linear combination of BALD and EPIG. While a combination approach is not necessarily a problem, the paper does not convincingly demonstrate that such a simple combination yields a significant improvement in most of the experiments. Moreover, even though the authors explain why $\beta=EIG$ is suggested, I still feel this choice to be rather heuristic and not theoretically grounded.

---

### Comment · Area_Chair_U8Rr · 2025-08-08
**Diverging Opinions in Reviews**

Dear Reviewers,

thx for engaging in discussions with the authors. Your opinions are diverging, so it would help my work if you could give a qualified statement during the AC-reviewer discussion period.

If you need further information from the authors, can you try to use the remaining time for clarifying questions or answering their comment?

Thx, AC

---

### Decision · Program_Chairs · 2025-09-17

**Decision:**

Accept (poster)

**Comment:**

The paper proposes HIPE, a hyperparameter-free initialization strategy for large-batch, few-shot Bayesian Optimization and Active Learning, combining BALD and EPIG in an information-theoretic framework. The method is simple, well-motivated, and easy to implement, addressing an underexplored but practically important problem.

The main point of disagreement among reviewers is novelty: while some consider the contribution incremental (a linear combination of known acquisitions), most acknowledge that identifying and automating this combination for hyperparameter initialization is new in this context. Reviewers also note the method’s strong practical appeal, solid empirical performance, and thorough presentation.

I examined the paper and find the experimental coverage sufficient to demonstrate the method’s merit. Although broader benchmarks and theoretical guarantees could strengthen the work, the majority of reviewers (including several strong accepts) find it interesting and novel enough for NeurIPS. The contribution is practically relevant, well-supported by experiments, and of interest to the NeurIPS community, with novelty concerns outweighed by its usefulness.